# Single-cell expression profiling reveals dynamic flux of cardiac stromal, vascular and immune cells in health and injury

**Nona Farbehi[1,2,3,4†], Ralph Patrick[1,2,5†], Aude Dorison[1,2], Munira Xaymardan[1,2,6], Vaibhao Janbandhu[1,2,5], Katharina Wystub-Lis[1], Joshua WK Ho[1,5], Robert E Nordon[2,4]\*, Richard P Harvey[1,2,7]\***

[1]Victor Chang Cardiac Research Institute, Darlinghurst, Australia; [2]Stem Cells Australia, Melbourne Brain Centre, University of Melbourne, Victoria, Australia; [3]Garvan Weizmann Centre for Cellular Genomics, Garvan Institute of Medical Research, Sydney, Australia; [4]Graduate School of Biomedical Engineering, UNSW Sydney, Kensington, Australia; [5]St. Vincent's Clinical School, UNSW Sydney, Kensington, Australia; [6]School of Dentistry, Faculty of Medicine and Health, University of Sydney, Westmead Hospital, Westmead, Australia; [7]School of Biotechnology and Biomolecular Science, UNSW Sydney, Kensington, Australia

**\*For correspondence:**
r.nordon@unsw.edu.au (REN);
r.harvey@victorchang.edu.au
(RPH)

[†]These authors contributed equally to this work

**Abstract** Besides cardiomyocytes (CM), the heart contains numerous interstitial cell types which play key roles in heart repair, regeneration and disease, including fibroblast, vascular and immune cells. However, a comprehensive understanding of this interactive cell community is lacking. We performed single-cell RNA-sequencing of the total non-CM fraction and enriched (*Pdgfra*-GFP[+]) fibroblast lineage cells from murine hearts at days 3 and 7 post-sham or myocardial infarction (MI) surgery. Clustering of >30,000 single cells identified >30 populations representing nine cell lineages, including a previously undescribed fibroblast lineage trajectory present in both sham and MI hearts leading to a uniquely activated cell state defined in part by a strong anti-WNT transcriptome signature. We also uncovered novel myofibroblast subtypes expressing either pro-fibrotic or anti-fibrotic signatures. Our data highlight non-linear dynamics in myeloid and fibroblast lineages after cardiac injury, and provide an entry point for deeper analysis of cardiac homeostasis, inflammation, fibrosis, repair and regeneration.
DOI: https://doi.org/10.7554/eLife.43882.001

## Introduction

Cardiovascular disease including myocardial infarction (MI) remains a leading cause of morbidity and mortality in the Western and developing worlds. After acute MI, millions of cardiomyocytes (CM) are lost by necrosis and apoptosis, and an initially adaptive collagen-rich scar is laid down to preserve chamber geometry and prevent rupture. The mammalian heart is regarded as being poorly regenerative as the long-term sequelae in virtually all etiologies of heart disease involve increased wall stiffness, reduced heart function and progression to heart failure. However, some inbred strains of mice show surprising cardiac reparative abilities (*Patterson et al., 2017*), and CM renewal and heart regeneration can be stimulated experimentally (*D'Uva et al., 2015*; *Mohamed et al., 2018*; *Srivastava and DeWitt, 2016*; *Wang et al., 2018*), garnering optimism that heart regeneration can be achieved in humans.

Cardiac chamber walls are composed of a complex, interdependent community of interstitial cells, including vascular, fibroblast, immune and neuronal cells, although how they interact in cardiac homeostasis, injury and repair, is relatively unexplored. In regenerative systems, connective tissues

**eLife digest** In our bodies, heart attacks lead to cell death and inflammation. This is then followed by a healing phase where the organ repairs itself. There are many types of heart cells, from muscle and pacemaker cells that help to create the beating motion, to so-called fibroblasts that act as a supporting network. Yet, it is still unclear how individual cells participate in the heart's response to injury.

All cells possess the same genetic information, but they turn on or off different genes depending on the specific tasks that they need to perform. Spotting which genes are activated in individual cells can therefore provide clues about their exact roles in the body. Until recently, technological limitations meant that this information was difficult to access, because it was only possible to capture the global response of a group of cells in a sample.

A new method called single-cell RNA sequencing is now allowing researchers to study the activities of many genes in thousands of individual cells at the same time. Here, Farbehi, Patrick et al. performed single-cell RNA sequencing on over 30,000 individual cells from healthy and injured mouse hearts. Computational approaches were then used to cluster cells into groups according to the activities of their genes.

The experiments identified over 30 distinct sub-types of cell, including several that were previously unknown. For example, a group of fibroblasts that express a gene called *Wif1* was discovered. Previous genetic studies have shown that *Wif1* is essential for the heart's response to injury. Further experiments by Farbehi, Patrick et al. indicated that this new sub-type of cells may control the timing of the different aspects of heart repair after damage.

Tens of millions of people around the world suffer from heart attacks and other heart diseases. Knowing how different types of heart cells participate in repair mechanisms may help to find new targets for drugs and other treatments.

DOI: https://doi.org/10.7554/eLife.43882.002

play key roles in defining positional information, and organizing tissue architecture and niche environments (*Nacu et al., 2013*; *Chan et al., 2013*; *Greicius et al., 2018*). Cardiac fibroblasts represent ~10% of all cardiac cells (*Pinto et al., 2016*) and are distributed throughout the cardiac interstitial, perivascular and sub-epicardial spaces, where they are proposed to have sentinel, paracrine, mechanical, extracellular matrix (ECM) and electrical functions (*Shinde and Frangogiannis, 2014*; *Tallquist and Molkentin, 2017*). After injury, inflammation is principally executed by polyfunctional monocytes (Mo) and macrophages (MΦ), and is necessary to protect against pathogens and autoimmunity, and to coordinate healing. Fibroblasts also participate in inflammation and phagocytosis and are the principal drivers of fibrotic repair (*Shinde and Frangogiannis, 2014*; *Gourdie et al., 2016*). In heart repair, timely resolution of inflammation is necessary for limiting fibrosis and enabling tissue replacement, while uncontrolled inflammation leads to increased fibrosis and chamber wall stiffening, poor electro-mechanical coupling, continued loss of CMs and worsening outcomes (*Mescher, 2017*; *Lai et al., 2017*; *Williams et al., 2018*).

The general principles of inflammation and fibrosis have been mapped in different organs, and the implementation of specific lineage-tracing tools has provided significant new insights into cardiac leukocyte and fibroblast origins and fate (*Tallquist and Molkentin, 2017*; *Williams et al., 2018*; *Fu et al., 2018*; *Ivey et al., 2018*; *Kanisicak et al., 2016*; *Moore-Morris et al., 2014*; *Ensan et al., 2016*; *Heidt et al., 2014*; *Molawi et al., 2014*; *Epelman et al., 2014*; *Plein et al., 2018*). However, controversies persist around nomenclature, defining markers, origins, heterogeneity and plasticity (*Tallquist and Molkentin, 2017*; *Epelman et al., 2015*; *Swirski and Nahrendorf, 2018*). Even the question of whether the transitions from quiescent to activated fibroblast, then to myofibroblast, should be seen as differentiation in the classical sense, degrees of a scalable and reversible continuum governed by the injury environment, or a branched dynamic network, is unresolved (*Tallquist and Molkentin, 2017*; *Ivey and Tallquist, 2016*; *Travers et al., 2016*).

One approach to a deeper understanding of cardiac population biology is through single-cell genomics, including single-cell RNA sequencing (scRNA-seq). Single-cell methods have the power to overcome the limitations of bulk cell analyses, where insights into complex cell system dynamics are

lost (*Tanay and Regev, 2017*). The rich data generated by single-cell methods allow new computational frameworks for inferring cell dynamics and causality, unencumbered by strict a priori notions of cell identity, hierarchy, trajectory and markers.

Here, we present the first comprehensive analysis of cellular lineage heterogeneity, dynamics and intercellular communication among immune and stromal (non-CM) cells in healthy and injured adult mouse hearts using scRNA-seq. Clustering analysis of >30,000 cells identified over 30 cell populations across the total non-CM fraction and enriched (*Pdgfra*-GFP⁺) fibroblast lineage cells. These populations comprised most of the known cell types and their dynamics after injury, as well as novel cell types and their intermediates. We describe a novel population of activated fibroblasts present in both sham and injured hearts expressing a strong anti-*Wingless-related integration site* (WNT) transcriptome signature, a putative pre-proliferative state, and three novel myofibroblast subtypes expressing pro-fibrotic or anti-fibrotic (including anti-WNT) signatures. We were also able to distinguish the major tissue-resident and infiltrating Mo/MΦ, and numerous minor populations. Overall, our data reveal dynamic, multi-dimensional lineage trajectories in the injured heart. This deep resource will provide novel insights into the inflammatory and fibrotic cascades in the injured mouse heart that may suggest novel molecular or cellular targets for enhancing heart repair and regeneration in man.

## Results

### Single-cell RNA-seq of total cardiac interstitial cell population

We performed single-cell expression profiling on the total cardiac interstitial cell population (TIP) using the 10x Genomics Chromium platform, from hearts of 8 weeks old male *Pdgfra*$^{GFP/+}$ mice at days 3 and 7 post-sham or MI surgery. To enrich for cells relevant to cardiac ischemic injury and repair, we isolated TIP cells from dissected ventricles and interventricular septum, excluding cells of the atria, *annulus fibrosus* and atrioventricular valves (*Figure 1—figure supplement 1A*).

Transcriptional profiles of 13,331 cells were captured after quality control filtering (sham: 5,723; MI-day 3: 3,875; MI-day 7: 3,733). To identify cells with distinct lineage identities and transcriptional states, we performed unbiased clustering on an aggregate of cells using the *Seurat* R package (*Butler et al., 2018*), with cell populations visualized in t-SNE dimensionality reduction plots (Materials and methods). For initial analyses, populations expressing markers of multiple lineages (hybrids) were removed; however, select examples are discussed in more detail below.

TIP cells were represented by a total of 24 populations and nine distinct cell lineages (*Figure 1A–D*; *Figure 1—figure supplement 1B–F*). Major cell types comprised fibroblasts/myofibroblasts (*Col1a1⁺Pdgfra⁺GFP⁺*), endothelial cells (ECs; *Kdr⁺Pecam1⁺*), mural cells (*Cspg4⁺Pdgfrb⁺*), Mo and MΦ (*Cd68⁺Itgam⁺*), dendritic-like (DC) cells (*Cd209a⁺Itgam⁺*), glial cells (*Plp1⁺Kcna1⁺*), B-cells (*Cd79a⁺Ms4a1⁺*), T-cells (*Cd3e⁺Cd3d⁺Lef1⁺*) and natural killer cells (NKCs; *Klrk1⁺Ccl5⁺*) (*Figure 1A–E*; *Figure 1—figure supplement 1D*; *Supplementary file 1*). New lineage markers were identified; for example, *Vtn*, encoding Vitronectin, was specifically expressed in mural cells, whereas *Kcna1*, encoding the potassium voltage-gated channel subfamily A member 1, was highly specific to glial cells (*Figure 1D* and *Figure 1—figure supplement 1D*).

Within the fibroblast lineage, we identified several sub-populations, each marked by expression of *Pdgfra*, *Pdgfra*-GFP, *Col1a1* and other canonical fiboblast markers (*Tallquist and Molkentin, 2017*; *Ivey and Tallquist, 2016*) (*Figure 1A–D*; *Figure 1—figure supplement 2A*). We describe these in more detail after enrichment below.

There were three major sub-populations of ECs (EC1, EC2, EC3), comprising vascular and lymphatic lineages (*Pecam1⁺Kdr⁺Ly6a⁺*) (*Figure 1D*; *Figure 1—figure supplement 2B*). The majority EC1 population expressed *Ly6a* (encoding SCA1) as well as the vascular transcription factor (TF) *Sox17*, and likely represents microvascular ECs. EC2 expressed canonical arterial endothelial markers such as *Bmx*, *Sema3g* and *Efnb2*, as well as TF genes *Sox17* and *Hey1* (*Figure 1—figure supplement 2B*), the latter acting downstream of NOTCH which is required for arterial EC fate. EC3 almost uniquely expressed venous EC marker *Nr2f2* (encoding COUPTFII) and Von Willebrand factor (*Vwf*), and a minority (~3%) expressed *Prox1* and *Lyve1* (*Figure 1—figure supplement 2B*), consistent with a lymphatic identity. A small number of ECs were cycling (Cyc; *Figure 1A–C*; *Figure 1—figure*

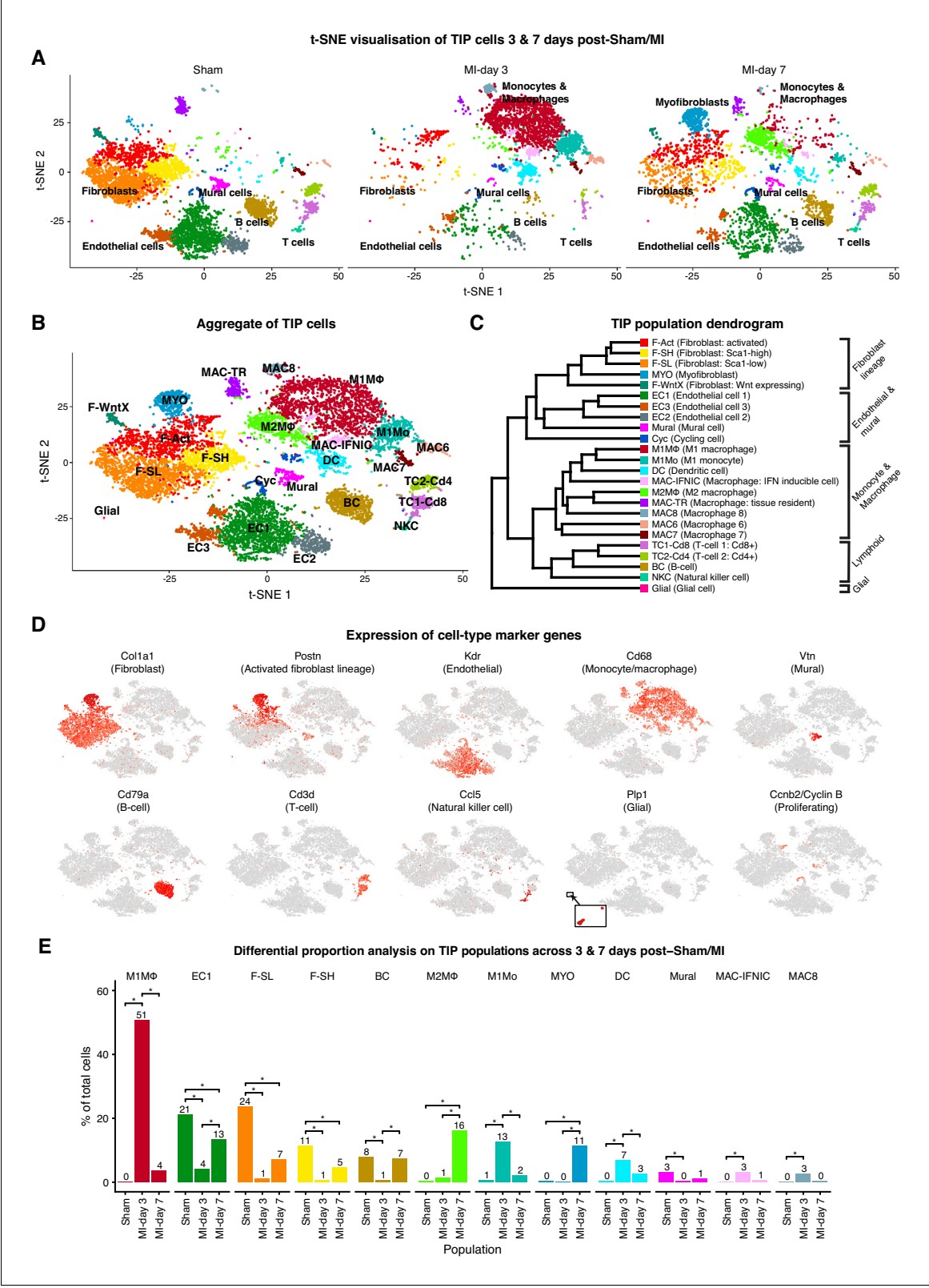

**Figure 1.** TIP scRNA-seq. (**A**) t-SNE plots showing detected lineages and sub-populations in TIP across conditions. (**B**) t-SNE plot of aggregate TIP cells with identified sub-populations. (**C**) Dendrogram of sub-populations according to average RNA expression. (**D**) Expression of select marker genes across TIP cells as visualized on t-SNE plots. (**E**) Cell population percentages across conditions determined to be significantly modulated according to Differential Proporation Analysis (DPA) (p<0.01).

*Figure 1 continued on next page*

*Figure 1 continued*

DOI: https://doi.org/10.7554/eLife.43882.003

The following source data and figure supplements are available for figure 1:

**Source data 1.** Source data for FACS quantifications summarized in *Figure 1—figure supplement 6D,E* and *Figure 1—figure supplement 7B,C*.
DOI: https://doi.org/10.7554/eLife.43882.012

**Figure supplement 1.** Experimental procedures, population proportions and gene expression characterstics of sub-populations within TIP scRNA-seq.
DOI: https://doi.org/10.7554/eLife.43882.004

**Figure supplement 2.** Gene expression data for TIP sub-populations.
DOI: https://doi.org/10.7554/eLife.43882.005

**Figure supplement 3.** Differential proportion analysis (DPA) procedure and evaluation.
DOI: https://doi.org/10.7554/eLife.43882.006

**Figure supplement 4.** Clustering of TIP scRNA-seq prior to removal of minor hybrid populations.
DOI: https://doi.org/10.7554/eLife.43882.007

**Figure supplement 5.** A typical workflow of sequential gating strategy for doublet exclusion is shown.
DOI: https://doi.org/10.7554/eLife.43882.008

**Figure supplement 6.** FACS data supporting F-EC population.
DOI: https://doi.org/10.7554/eLife.43882.009

**Figure supplement 7.** FACS data supporting the M2 MΦ -EC population.
DOI: https://doi.org/10.7554/eLife.43882.010

**Figure supplement 8.** Atrial Fibrilation (AF) associated genes in TIP.
DOI: https://doi.org/10.7554/eLife.43882.011

*supplement 2C*). Our EC data are broadly consistent with recently published single-cell data (*Zhao et al., 2018*).

Among lymphocytes, a single B-cell population (BC) expressed *Cd79a, Ms4a1* and *Ly6d* (*Figure 1D*; *Figure 1—figure supplement 2D*). T-cell sub-populations TC1-Cd8 (*Cd8a*$^+$) and TC2-Cd4 (*Cd4*$^+$*Lef1*$^+$) likely represent cytotoxic and helper T-cells, respectively. NKCs exclusively expressed *Klrk1* and upregulated *Lck, Ccl5* and *Ctsw* (*Figure 1—figure supplement 2D*).

## Differential proportion analysis for detecting cell population dynamics

We observed major injury-induced cellular responses and flux after MI, including expansion of Mo/MΦ populations at MI-day 3, as well as myofibroblasts and an additional MΦ population at MI-day 7 (*Figure 1A*; *Figure 1—figure supplement 1B,C*). To analyze whether changes in the proportion of populations were greater than expected by chance, we developed a novel permutation-based statistical test (differential proportion analysis; DPA) that considered sources of variation which could arise from experimental procedures (such as differing cell numbers and cell-type capture bias) or in silico analysis (cluster assignment accuracy) (Materials and methods; *Figure 1—figure supplement 3A–H*). DPA identified 12 populations showing significant (p<0.01) flux between conditions (*Figure 1E*; *Supplementary file 2*); for example, the fibroblast sub-populations F-SL and F-SH (see below) decreased sharply in proportion at MI-day 3, while M1 and M2 MΦ populations expanded at days 3 and 7 after MI, respectively.

## Monocyte/macrophage cell identity and dynamics

Cardiac tissue-resident MΦ originate from CX3CR1$^+$ progenitors in the yolk sac and Mo from fetal liver and post-natal bone marrow (*Ensan et al., 2016*), and have roles in immunity, coronary artery and pacemaker development, and heart regeneration (*Lavine et al., 2014*; *Hulsmans et al., 2017*). Resident MΦ are long lived and self-renewing (*Epelman et al., 2014*; *Bajpai et al., 2018*), although some are supplanted by blood-derived Mo with age or injury (*Heidt et al., 2014*; *Molawi et al., 2014*; *Dick et al., 2019*). MI triggers a biphasic cascade of inflammation and remodeling, with the acute phase involving early influx of neutrophils and CCR2$^+$LY6C2$^{high}$ pro-inflammatory M1 Mo/MΦ, which phagocytose debris and secrete pro-inflammatory factors IL-1β, IL-6 and TNFα to amplify the inflammatory response (*Swirski and Nahrendorf, 2018*). The repair phase begins around MI-day 3 when non-classical LY6C2$^-$F4/80$^{high}$ M2 MΦ accumulate and secrete anti-inflammatory cytokines such as Il-10 and TGF-β, and stimulate angiogenesis (*Epelman et al., 2015*; *Swirski and Nahrendorf, 2018*).

In sham hearts, we identified cardiac tissue-resident MΦ with the signature $Cx3cr1^{high}Adgre1(F4/80)^{high}H2-Aa(MHC-II)^{+}Itgam(CD11b)^{low}Ly6c2^{low}Ccr2^{-}$ (MAC-TR; *Figure 1A,B*; *Figure 2A–D*) (*Ensan et al., 2016*; *Epelman et al., 2014*; *Swirski and Nahrendorf, 2018*; *Lavine et al., 2014*; *Lavine et al., 2018*). Recent work using flow cytometry and scRNA-seq has delineated several sub-sets of cardiac tissue-resident MΦ, including a pro-regenerative population with the signature $TIMD4^{+}LYVE1^{+}MHC-II^{low}CCR2^{-}$, that self-renew and are not replaced by blood monocytes even after injury (*Dick et al., 2019*). We could discern this same population at the scRNA-seq level as a subset within MAC-TR, which persisted after injury (*Figure 2—figure supplement 1A*; *Figure 2B*). The additional major subset of CCR2$^{-}$ tissue-resident MΦ (*Dick et al., 2019*) could also be recognised at the scRNA-seq level as the $Timd4^{-}Lyve1^{-}H2-Aa(MHC-II)^{high}Ccr2^{-}$ subset of MAC-TR – this population has been shown to have a low monocyte dependence during homeostasis but is almost fully replaced by monocytes after MI (*Dick et al., 2019*).

Among other minor resident Mo/MΦ populations detected, the most abundant (pale green cells in *Figure 1A,B*) clustered with the M2 MΦ present at MI-day 7. In fact, all minor Mo/MΦ populations in sham hearts aligned with adult monocyte-derived Mo/MΦ populations which influx after MI (*Figure 1A,B*), consistent with recent findings (*Dick et al., 2019*). A prominent B-cell, and minor DC, T- and NK cell populations were also present in sham hearts. These populations may represent a mixture of resident cells and those involved in homeostatic immunosurveillance (*Lavine et al., 2018*), although we cannot exclude a response to sham operation.

At MI-day 3, a major influx population was identified as classical blood-derived M1 Mo, based on the signature $Adgre1(F4/80)^{+}Itgam(CD11b)^{+}Fcgr1(CD64)^{+}Ly6c2^{high}Ccr2^{high}H2-Aa(MHC-II)^{low}$ (M1Mo; *Figure 2A–C*) (*Epelman et al., 2015*; *Swirski and Nahrendorf, 2018*). Differentially expressed genes showed Gene Ontology (GO) term over-representation for cell migration, inflammation and T cell activation (*Figure 2D*). In FACS, Mo are distinguished from MΦ by having lower size and granularity, and lower levels of MΦ markers including *Adgre(F4/80)*, *Itgam (CD11b)* and *H2-Aa(MHC-II)* (*Bajpai et al., 2018*; *Hilgendorf et al., 2014*). M1Mo identified at MI-day 3 were also low or negative for other MΦ markers *Siglec1*, *Mrc1*, *Maf*, *Trem2* and *Mertk*, the latter involved in phagocytosis (*Figure 2—figure supplement 1A*) (*Bajpai et al., 2018*), and C1 complement genes *C1qa*, *b* and *c* (*Figure 2C*), which are involved (in addition to complement fixation) in recruitment of new inflammatory cells and protection against autoimmunity (*Emmens et al., 2017*; *Thielens et al., 2017*).

The more abundant population at MI-day 3 was identified as classical Mo-derived M1 MΦ based on the signature $Ccr2^{high}Adgre1(F4/80)^{+}Ly6c2^{+}H2-Aa(MHC-II)^{+}$ (M1MΦ; *Figure 1A,B*; *Figure 2A–D*). This assignment was supported by expression of the additional MΦ markers cited above, including *Mertk* and *C1q*, and hierarchical clustering, which showed M1MΦ to be most closely related to M1Mo (*Figure 1C*), as for human cognates (*Bajpai et al., 2018*). Differentially expressed genes showed GO term over-representation for leukocyte migration and responses to interleukin-1 (*Figure 2D*).

The most prominent population at MI-day 7 was identified as non-classical M2 MΦ involved in inflammation resolution and repair, with the signature $Ccr2^{high}Adgre1(F4/80)^{+}H2-Aa(MHC-II)^{high}Ly6c2^{-}$ (M2MΦ; *Figure 1A,B*; *Figure 2A–D*) (*Epelman et al., 2015*; *Swirski and Nahrendorf, 2018*). Differentially expressed genes showed GO term over-representation for antigen presentation via MHC class II (*Figure 2D*). As expected, the non-classical M2 MΦ population increased late during injury repair from <2% of TIP in sham and MI-day 3 hearts, to 16% at MI-day 7. Interestingly, the population dendrogram showed that M2MΦ were most closely related to MAC-TR (*Figure 1C*), and similarities between resident and subsets of infiltrating Mo/MΦ have been discerned recently using single-cell methods (*Dick et al., 2019*). Both MAC-TR and M2MΦ expressed *Cx3cr1*, often used to define tissue-resident MΦ (*Figure 2B*), and both upregulated pro-regenerative genes *Igf1* (*Figure 2B*) and *Pdgfb/c* (*Figure 2—figure supplement 1A*). The majority of M2MΦ were $Ccr2^{high}$ (important for migration); however, a minor sub-populaion was $Ccr2^{low}$ (arrows, *Figure 2B*) and these expressed the highest levels of *Igf1* and lower levels of *MHC-II* (*Figure 2—figure supplement 1B*). In this sense they are similar to the CCR2$^{-}$MHC-II$^{low}$ subset of tissue-resident MΦ which appear to be yolk sac-derived (*Dick et al., 2019*; *Leid et al., 2016*), and which play a role through expression of IGF1 and IGF2 in remodeling of the fetal coronary vascular plexus (*Leid et al., 2016*). However, whether in the adult post-MI heart they represent persisting resident cells or an infiltrating population that has matured into a MAC-TR-like MΦ state will require lineage mapping approaches.

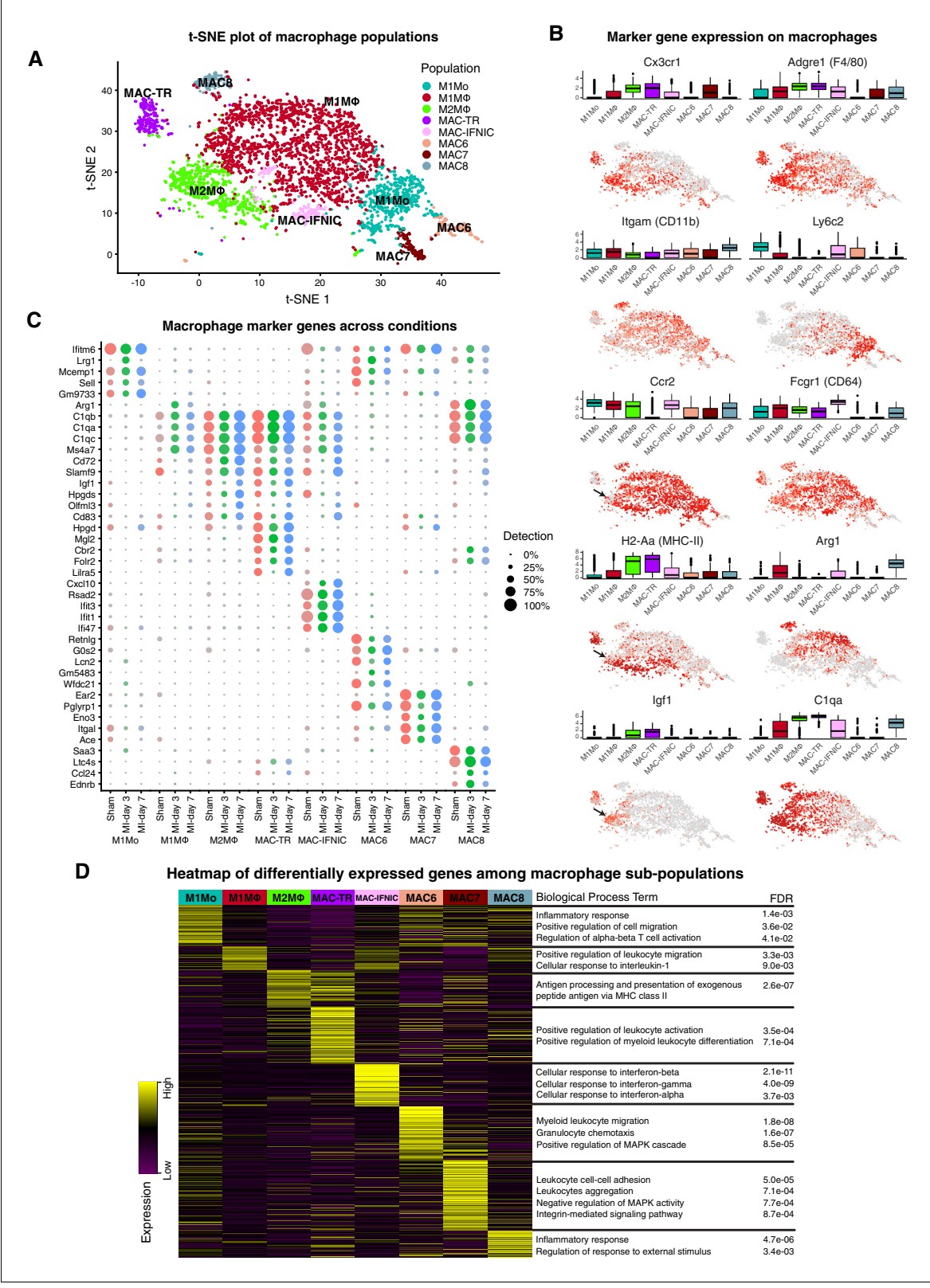

**Figure 2.** Cardiac Mo/MΦ populations. (**A**) t-SNE plot showing extracted Mo/MΦ populations. (**B**) Expression of select immune cell markers as visualized in box-plots and t-SNE plots. Arrows indicate Ccr2[low] sub-population of M2MΦ. (**C**) Dot-plot of top upregulated genes for each Mo/MΦ population where color indicates experimental conditions. (**D**) Heatmap of differentially expressed genes between Mo/MΦ populations with representative significant GO Biological Process terms.

*Figure 2 continued on next page*

*Figure 2 continued*

DOI: https://doi.org/10.7554/eLife.43882.013

The following figure supplement is available for figure 2:

**Figure supplement 1.** Mo and MΦ marker genes and Diffusion Map analysis.

DOI: https://doi.org/10.7554/eLife.43882.014

*Il10*, associated with the anti-inflammatory functions of M2MΦ, was expressed in only few cells in our dataset and may be at the limit of detection (*Figure 2—figure supplement 1A*). Expression of mouse genes encoding cognates of human CD14 and CD16/FCGR3, previously used to define classical, non-classical and intermediate Mo in human blood (*Villani et al., 2017*), did not help to discrimate the above Mo/MΦ populations, nor did new markers recently highlighted from CyTOF analysis (*Williams et al., 2018*). Moreover, the M2MΦ marker *Arg1* (encoding Arginase 1) was more lowly expressed in M2MΦ described here than in M1MΦ, consistent with findings that ARG1 does not always mark M2 cells (*Jablonski et al., 2015*). Neither the M2 MΦ, nor any other myeloid population, expressed *Col1a* genes, likely precluding the presence of myeloid-derived fibroblasts (*Duerrschmid et al., 2015*).

*Diffusion Map* (*Angerer et al., 2016*) analysis applied to model possible temporal (pseudotime) changes in major Mo/MΦ populations (*Figure 2—figure supplement 1C*) revealed a continuum of states resolved into a trajectory from early infiltrating M1Mo (left) and inflammatory M1MΦ (centre), to the late peaking M2MΦ (right), similar to a recent scRNA-seq study (*Dick et al., 2019*) and consistent with the current model in which M1 Mo differentiate into M2 cells in situ (*Lavine et al., 2018*; *Hilgendorf et al., 2014*). The *Diffusion Map* also demonstrated the convergence of M2MΦ present at MI-day 7 with tissue-resident MΦ (MAC-TR) in sham hearts, a relationship reflected in the population dendrogram (*Figure 1C*; see Discussion).

The minor myeloid populations also showed different expression profiles and dynamics (*Figure 1A–C,E*; *Figure 2A–D*). For example, MAC6 showed upregulation of granulocyte markers including *S100a9* and *Csf3r* (*Supplementary file 3*), with sub-populations expressing markers of neutrophils (*Ly6g*) and eosinophils (*Siglecf*) (*Figure 2—figure supplement 1A*). MAC-IFNIC cells showed strong upregulation of interferon (IFN)-induced genes including *Ifit3*, *Ifit1* and *Cxcl10* (*Figure 2C*), consistent with GO term analysis implicating responses to IFN α, β, and γ (*Figure 2D*). These cells appear to arise from $Ccr2^+$ MΦ as opposed to monocytes (*Dick et al., 2019*), and likely correspond to the recently described inflammatory MΦ subtype that has negative effects on heart repair after MI through promotion of inflammatory cell types, and cytokine and chemokine expression (*King et al., 2017*).

## Cell-cell communication analysis in TIP

We constructed cell-cell communication networks with weighted edges reflecting expression fold-changes of ligands and receptors in source and target populations, respectively (Materials and methods). Ligand-receptor interactions were derived from a curated map of human ligand-receptor pairs (*Ramilowski et al., 2015*) with mouse-specific weights added after reference to the STRING database (*Szklarczyk et al., 2017*). Based on permutation testing of randomized network connections, 91 cell-cell relationships with weighted paths higher than expected by chance ($P_{adj}$ <0.01) were identified (*Figure 3A–B*). Myofibroblasts (MYO) and MΦ populations M1MΦ and MAC-8 exhibited the largest number of outbound connections, with MYO having the highest weight. ECs by far showed the largest number and weighting of significant inbound connections. Strikingly, fibroblast populations (F-SH, F-SL, F-Act and F-WntX) appeared to communicate exclusively with vascular (ECs and mural) and glial cells. In line with this result, immunofluorescence analysis of sham and MI-day 3 hearts showed that *Pdgfra*-GFP$^+$ fibroblasts were observed in close spatial relationships or direct contact with CD31$^+$ endothelial cells (*Figure 3—figure supplement 1A–B*).

Top-weighted interactions involving MYO were driven mostly by ECM-associated ligands including COL1a1, COL1a2, Fibronectin 1, Pleiotrophin, COL3a1, COL5a2, Biglycan, Metalloprotease inhibitor one and COL5a1, that can engage with receptors expressed in numerous populations (*Figure 3C*). The minority glial cell population expressed canonical neuronal glia markers, including *Plp1*, *Prnp* and *Gfra3* (*Figure 1—figure supplement 1D*), and likely support cardiac sympathetic

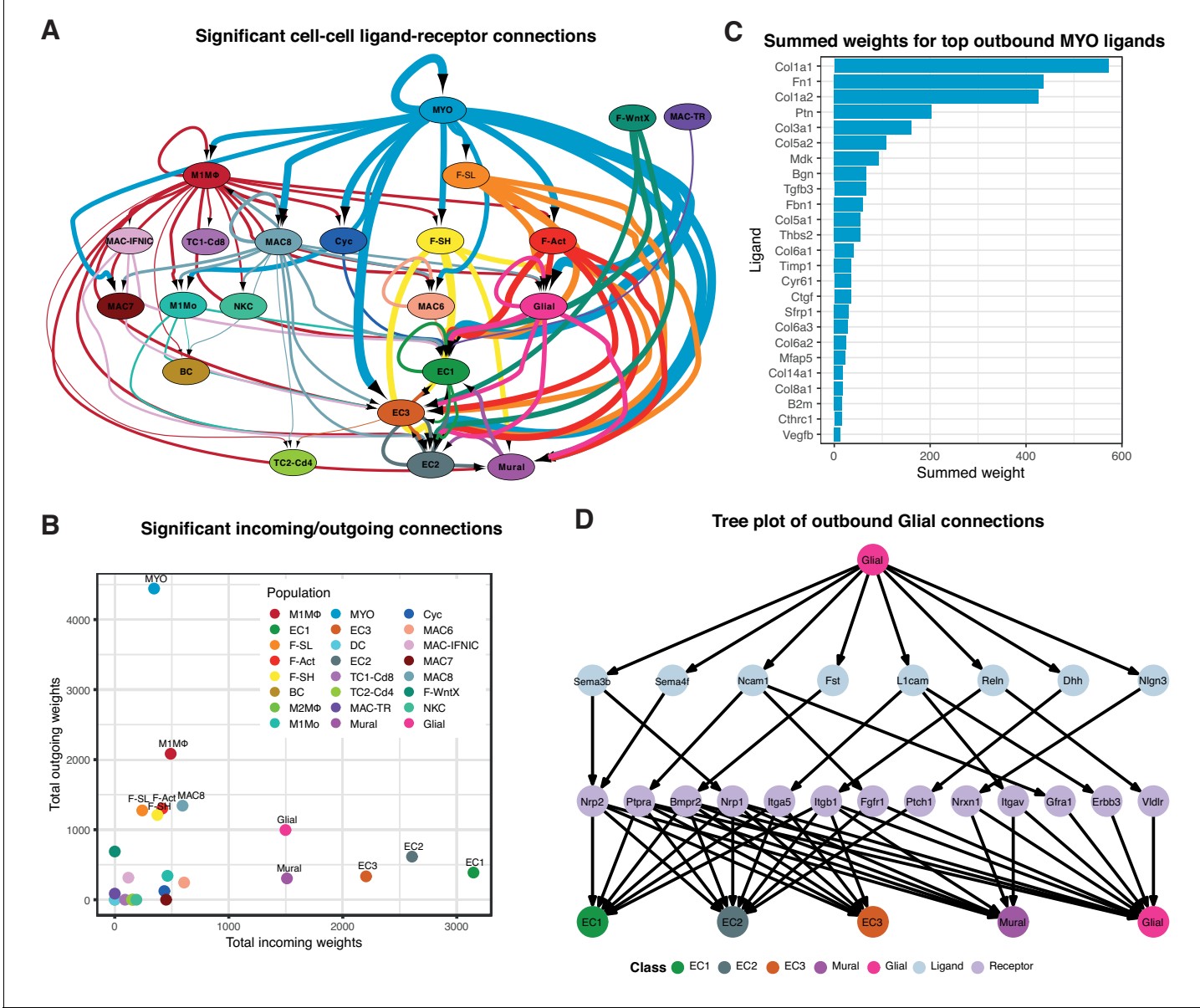

**Figure 3.** Cell-cell ligand-receptor network analysis. (A) Hierarchical network diagram of significant cell-cell interaction pathways. Arrows and edge color indicates direction (ligand:receptor) and edge thickness indicates the sum of weighted paths between populations. (B) Comparison of total incoming path weights vs total outgoing path weights across populations. (C) Summed ligand weights across souce ligand and receptor target paths for top ligands in MYO. (D) Tree plot showing outgoing connections from the Glial cells. Top node refers to source population, second layer to ligands, third layer to receptors and leaf nodes represent target populations.

DOI: https://doi.org/10.7554/eLife.43882.015

The following figure supplement is available for figure 3:

**Figure supplement 1.** *Pdgfra*-GFP$^+$ cells localization in healthy and diseased hearts.

DOI: https://doi.org/10.7554/eLife.43882.016

nerves essential for cardiac regeneration in neonates (*Mahmoud et al., 2015*). Glial cells also appeared to communicate with the three EC populations and mural cells (*Figure 3A*), consistent with the phenomenon of neurovascular congruence in the cardiac sympathetic plexus (*Stubbs et al., 2009*). In support of this, we detected eight ligands highly specific to glial cells (expressed in <5% of other TIP cells) including *Dhh* (Desert Hedgehog) and Semaphorin genes *Sema3b* and *Sema4f* (*Figure 3D*), involved in both neural and angiogenic development (*Gamboa et al., 2017*). Thus,

these maps suggest the extent, directionality and complexity of interactions between cardiac cell types in homeostasis and injury.

## Hybrid populations in TIP

We detected five minor populations expressing markers of two lineages (*Figure 1—figure supplement 4A–D*). Such 'hybrid' cells may betray trans-differentiation events or doublets in proximity that are resistant to the conditions of dissociation. Microdroplet microfluidics platforms are also known to generate a significant number of doublets (*Zheng et al., 2017a*); thus, the provenance of hybrid cells requires independent validation.

ECs are highly plastic and endothelial-to-mesenchyme transition (EndMT) has been reported to generate fibroblasts after cardiac injury (*Kovacic et al., 2019*). Conversely, cardiac fibroblasts have been observed to transdifferentiate to ECs (*Ubil et al., 2014*), albeit that this has been disputed (*He et al., 2017*). The F-EC hybrid population co-expressed markers of fibroblasts and ECs, and segregated with other fibroblast populations (*Figure 1—figure supplement 4A–C*). To explore this population, we isolated interstitial cells from dissected ventricles of $Pdgfra^{GFP/+}$ mice after sham or MI surgery, and asked whether we could detect GFP$^+$CD31$^+$ cells using flow cytometry and a stringent gating strategy that excluded cell aggregates (Materials and methods; *Figure 1—figure supplement 5A–D*). We detected 2.4 ± 0.28% of GFP$^+$CD31$^+$ cells in sham hearts, 1.51 ± 0.26% in MI-day 7 hearts, and none in controls (*Figure 1—figure supplement 6A–E*) - thus, while double positive cells were found, they did not appear responsive to injury.

An ability of Mo/MΦ to transdifferentiate into endothelial-like cells in different settings has been documented in vitro and in vivo, and has therapeutic implications (*Das et al., 2015*), although a natural plasticity in Mo toward an endothelial cell fate in vivo does not have strong support (*Basile and Yoder, 2014*). The M2MΦ-EC hybrid population co-expressed markers of ECs ($Kdr^+Pecam1^+Sox17^+$ $Efnb^+Mcam^+$) and M2MΦ ($Ccr2^{high}Adgre1[F4/80]^+H2-Aa[MHC-II]^{high}Cx3cr1^+Mrc1[CD206]^+Ly6\ c2^-$), and segregated with M2MΦ (*Figure 1—figure supplement 4A,B*). Flow revealed 0.56 ± 0.02% single live CD31$^+$CD45$^+$ cells in sham-day 7 hearts, increasing to 4.04 ± 1.03% in MI-day 7 hearts, demonstrating an increase in injury (*Figure 1—figure supplement 7A–C*). Among these, 35.67 ± 3.01% were F4/80$^+$CD206$^+$ (a signature of M1 and M2 MΦ) in sham hearts, increasing to 60.03 ± 4.60% in MI-day 7 hearts. It is well known that the expression of EC markers on the surface of bone-marrow-derived cells is insufficient to define them as ECs, although they can be angiogenesis promoting cells (*Basile and Yoder, 2014*). While these data do not exclude the possibility of doublet formation in our scRNA-seq experiments, they support the existence of distinct F-EC and M2MΦ-EC populations with hybrid qualities and different responses to injury. These warrant further investigation.

## Single-cell RNA-seq of the *Pdgfra*-GFP$^+$ cardiac fibroblast lineage

A major subset of fibroblasts in the uninjured adult murine heart express the cell surface stem/progenitor cell markers SCA1 and/or PDGFRα (*Kanisicak et al., 2016*; *Asli et al., 2017*; *Chong et al., 2011*; *Noseda et al., 2015*). However, when fibroblasts differentiate into MYO, they reduce these markers and express fibrogenic (e.g. Periostin; POSTN) and/or contractile (e.g. αSmooth Muscle Actin; αSMA) proteins (*Fu et al., 2018*). To circumvent the dominance of immune cells in TIP following MI, which dilute out other cell populations, and to focus on fibroblast sub-populations (*Figure 1A*), we performed single-cell expression profiling on PDGFRα$^+$CD31$^-$ cardiac interstitial cells at days 3 and 7 post-sham or MI. GFP fluorescence from $Pdgfra^{GFP/+}$ mice was used as a surrogate lineage tracing tool and enabled us to capture both GFP$^{high}$ fibroblasts as well as their derivatives in MI mice, including MYO (*Asli et al., 2017*). We sorted for GFP$^+$CD31$^-$ cells (*Figure 4—figure supplement 1A*), although did not use SCA1 as an index marker so as to capture the substantial *Pdgfra*-GFP$^+$ fibroblast population that is negative or low for SCA1 expression (*Figure 4—figure supplement 1B*).

We performed unbiased clustering on an aggregate of the 16,787 cells (Materials and methods), identifying 11 sub-populations (*Figure 4A–D*; *Figure 4—figure supplement 1C–F*). The two sham conditions showed high concordance (*Figure 4—figure supplement 1E,F*) and are displayed merged (*Figure 4A,B*) unless indicated. All populations showed expression of canonical fibroblast markers *Pdgfra*, *GFP*, *Ddr2* and *Col1a1*, albeit at varying proportions and levels (*Figure 4E*), and major changes in cell proportions were seen between conditions (*Figure 4D*). Here, we refer to

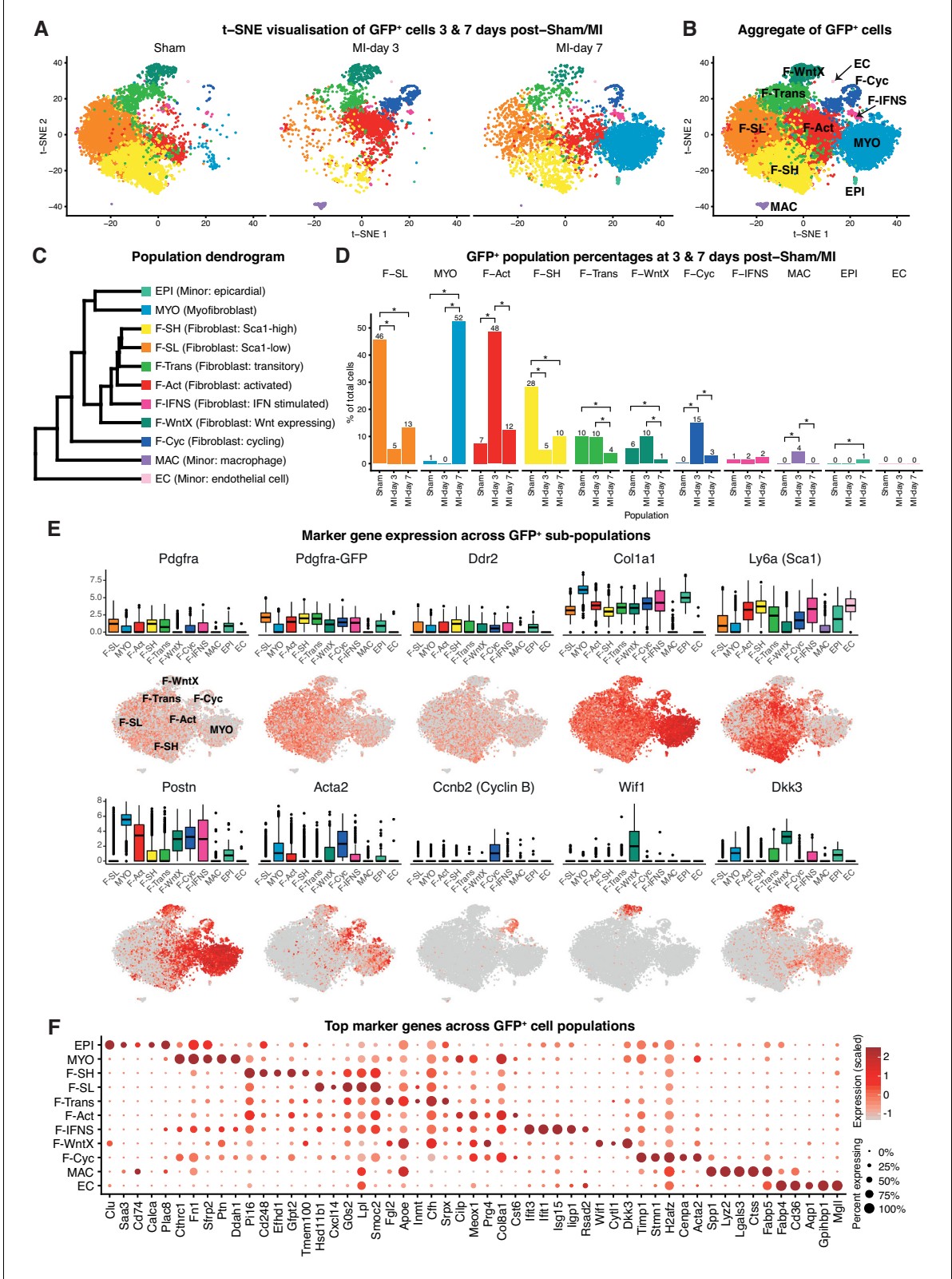

**Figure 4.** *Pdgfa*-GFP[+] scRNA-seq. (**A**) t-SNE plot of GFP[+] cells separated according to experimental condition (sham, MI-day 3, MI-day 7). (**B**) t-SNE plot showing aggregate of GFP[+] cells across conditions. (**C**) Dendrogram of populations determined by average RNA expression in populations. (**D**) Percentages of cells in each population according to experimental condition. Stars indicate significant change across conditions according to DPA

*Figure 4 continued on next page*

*Figure 4 continued*

(p<0.01). (E) Expression of select genes in different populations as visualized in box and t-SNE plots. (F) Dot-plot of top five upregulated genes for each population where color indicates strength of expression and size of dot represents percentage of cells expressing the gene.

DOI: https://doi.org/10.7554/eLife.43882.017

The following source data and figure supplements are available for figure 4:

**Source data 1.** Source data for quantification of colony counts summarized in *Figure 4—figure supplement 2E*.

DOI: https://doi.org/10.7554/eLife.43882.023

**Figure supplement 1.** FACS profiles and scRNA-seq analysis for GFP[+]/CD31[-] cells.

DOI: https://doi.org/10.7554/eLife.43882.018

**Figure supplement 2.** Comparison of GFP[+] populations with FACS-sorted S[+]P[+] cells.

DOI: https://doi.org/10.7554/eLife.43882.019

**Figure supplement 3.** Expression of selected marker genes across GFP[+] populations visualized in box and t-SNE plots.

DOI: https://doi.org/10.7554/eLife.43882.020

**Figure supplement 4.** Population-specific expression of transcription factors marking fibroblast and myofibroblast sub-populations.

DOI: https://doi.org/10.7554/eLife.43882.021

**Figure supplement 5.** Comparisons with *Skelly et al. (2018)* and *Gladka et al. (2018)* scRNA-seq data-sets.

DOI: https://doi.org/10.7554/eLife.43882.022

'activated fibroblasts' and myofibroblasts (MYO) as distinct cell entities, without prejudice about their stability, origin, fate or contractile nature.

In sham conditions, two major fibroblast populations could be distinguished on the basis of scRNA-seq. We termed these Fibroblast-Sca1-high (F-SH) and Fibroblast-Sca1-low (F-SL), as the highest upregulated gene in F-SH was *Ly6a*(*Sca1*) (*Figure 4E*; *Supplementary file 4*). F-SH contained the highest frequency of *Pdgfra* and *Ly6a*(*Sca1)*-expressing cells and likely corresponds to the PDGFRα[+]SCA1[+] (S[+]P[+]) population previously defined by FACS (*Pinto et al., 2016*; *Chong et al., 2011*) (see also *Figure 4—figure supplement 1A*) and enriched in cardiac colony-forming mesenchymal stromal cell (MSC)-like cells (cCFU-F), which show multi-lineage differentiation and self-renewal in vitro (*Chong et al., 2011*; *Noseda et al., 2015*). In order to confirm the relationship between F-SH and S[+]P[+], we performed deeper scRNA-seq on 103 FACS-purified S[+]P[+] cells from uninjured wild-type mice using the *Fluidigm* platform and predicted cell identity using an *iterative Random Forest* (iRF) classifier (*Basu et al., 2018*) trained on populations defined in our GFP[+] experiments in sham conditions using the *Chromium* platform (Materials and methods; *Figure 4—figure supplement 2A*). Approximately 60% of single S[+]P[+] cells analyzed by *Fluidigm* were predicted to correspond to the F-SH population (*Figure 4—figure supplement 2B*), compared to <30% among total sham GFP[+] cells (*Figure 4D*), indicating that S[+]P[+] cells are significantly over-represented in F-SH cells (Fisher's exact test, p=8.13e-11). We previously showed that cCFU-F are enriched in the S[+]P[+] population (*Chong et al., 2011*; *Noseda et al., 2015*). THY1/CD90 is a recognised MSC marker, and *Thy1* was upregulated in F-SH with high significance (p=4.48e-176; *Figure 4—figure supplement 2C*). Furthermore, FACS-isolated S[+]*Pdgfra*-GFP[+]CD90.2[high] cells isolated from healthy hearts showed a ~ 6 fold enrichment in cCFU-F compared to S[+]*Pdgfra*-GFP[+]CD90.2[low] cells (*Figure 4—figure supplement 2D,E*). Together, these results show that the F-SH population contains a subset of cells expressing *Pdgfra*, *Ly6a*(*Sca1*) and *Thy1*(*Cd90*) that is enriched in cCFU-F, highlighting the distinct expression signatures and functional properties of F-SH and F-SL.

We calculated differentially expressed (DE) genes between F-SL and F-SH in sham conditions (*Supplementary file 5*). F-SH was characterized by over-representation of genes involved in the biological process (BP) *cell adhesion*, which included cell surface receptor genes *Ackr3*(*Cxcr-7*), *Thy1* (*Cd90*), *Axl* and *Cd34*. In contrast, F-SL was characterized by GO BP terms *signaling* and *signal transduction* (*Supplementary file 6*). Within the *signal transduction* category, protein localization prediction with *LocTree3* (*Goldberg et al., 2014*) indicated an over-represented majority (19/28) of secreted proteins (Fisher's exact test, p=0.03), with 10/19 identified as ligands, including APOE, BMP4 and ADM. Thus, F-SL, a major sub-division of fibroblasts, has a unique secretory phenotype distinct from that in F-SH, which is enriched in MSC-like colony forming cells.

## Novel *Pdgfra*-GFP⁺ fibroblast populations

We identified two previously unstudied GFP⁺ fibroblast populations termed Fibroblast - Wnt expressing (F-WntX) and Fibroblast - transitory (F-Trans). These were present in both sham and MI hearts, although had diminished significantly by MI-day 7 (DPA; p<0.01; *Figure 4D*). In F-WntX, differential gene expression analysis showed that the top upregulated gene was *Wif1*, encoding a secreted canonical WNT pathway inhibitor essential for cardiac repair after MI (*Meyer et al., 2017*). WIF1 can also antagonize Connective Tissue Growth Factor (CTGF) signaling (*Surmann-Schmitt et al., 2012*), which plays a supportive role in cardiac fibrosis (*Travers et al., 2016*). *Wif1* was almost uniquely expressed in F-WntX in all conditions (*Figure 4E*). A single cell in the Fluidigm data corresponded to the F-Wntx population (*Figure 4—figure supplement 2B*). Multiple other WNT pathway-related genes were upregulated in F-WntX encoding WNT ligands (WNT5a, WNT16), soluble WNT antagonists (DKK3, SFRP2), membrane-bound WNT receptor (FRZB) and AXIN2, a component of the β-catenin destruction complex (see schematic in *Figure 5A*; *Figure 4—figure supplement 3*). F-WntX also showed upregulated *Fmod*, which inhibits fibrillogenesis and sequesters pro-fibrotic factor TGF-β within ECM (*Zheng et al., 2014*; *Zheng et al., 2017b*) (*Supplementary file 4*). Overall, this signature suggests an anti-WNT, anti-CTGF and anti-TGF-β extracellular and intracellular signaling milieu for F-WntX cells. F-WntX cells expressed *Postn*(*Periostin*), *Acta2*(*αSMA*), *Tagln* (*Transgelin*) and *Scx*(*Scleraxis*), in both sham and MI conditions (*Figure 5B*; *Figure 5—figure supplement 1A*), suggesting an activated state even in the absence of injury (*Tallquist and Molkentin, 2017*). The adjacent cluster, F-Trans, did not express activation markers, nor the WNT signature identified in F-WntX, or other uniquely identifying markers; however, cell trajectory analysis, described below, allowed us to assign F-Trans as a transitionary population between F-WntX and F-SL fibroblasts.

We examined DE and GO BP terms in F-WntX compared with F-SL and F-SH combined. Notably, *negative regulation of Wnt signaling pathway* was over-represented in DE genes for F-WntX (*Figure 5C*), driven by WNT pathway antagonist genes *Wif1*, *Dkk3*, *Frzb* and *Sfrp2*, discussed above, as well as *Apoe*, *Nkd1* and *Wwtr1*, also known to interact with the WNT pathway. BP terms related to negative regulation of development/differentiation, extracellular matrix (ECM) organization and signaling, were also significant. ECM organization terms were over-represented in the adjacent F-Trans (*Supplementary file 6*), involving genes also upregulated in F-WntX (e.g. *Eln*[*Elastin*], *Vit*[*Vitrin*] and *Mfap4*), and others upregulated in F-Trans but not F-WntX, including collagen genes (*Col1a1*, *Col3a1* and *Col14a1*) and *Fbln1*(*Fibrilin-1*).

The top GO BP term in F-WntX was *regulation of cell proliferation* (*Figure 5C*); however, F-WntX did not express cell cycle markers under any condition. Analysis of Molecular Function (MF) terms revealed over-representation of *signaling receptor binding* and *growth factor binding* (*Figure 5—figure supplement 1B*), overlapping significantly with *regulation of cell proliferation* (*Figure 5—figure supplement 1C,D*; Fisher's exact test, p<1e-03). In this latter term, there were several cytokine and chemokine genes, including *Pdgfa*, *Tgfb3*, *Ptn*, *Ccl19* and *Cxcl12*, some of which bind receptors that were down-regulated in F-WntX, strongly suggesting paracrine functions related to their expression in F-WntX (*Figure 5—figure supplement 1E*). A paracrine function for F-WntX was supported by our ligand-receptor analysis, which indicated that F-WntX cells communicate most significantly with ECs (*Figure 3A*). Analysis of top upregulated ligands in F-WntX connecting to receptors in ECs (*Figure 5D*) identified several factors such as *Ptn*(*pleiotrophin*), *Myoc*(*myocilin*) and *Timp3*(*TIMP metallopeptidase inhibitor 3*). Here again, the corresponding receptor was expressed in ECs but down-regulated in F-WntX (*Figure 5E*).

## Localization and composition of WIF1⁺ cells

To explore the location of F-WntX cells and their behaviour after injury, we examined the expression of WIF1 protein in *Pdgfra*-GFP sham and MI hearts by immunofluorescence (IF), after first confirming that our chosen antibody detected known sites of *Wif1* expression in E14.5 embryos (*Figure 6—figure supplement 1*). Interestingly, we detected WIF1 protein only in the infarct border zone at MI-day 3, but not in sham hearts or at MI-days 1 or 7 (*Figure 6A–C*, *Figure 6—figure supplement 2*). In cardiac cells (and some embryonic cells) WIF1 staining was perinuclear, and we demonstrated co-expression of WIF1 and the golgi marker GM130 (*Figure 6D*), consistent with WIF1 being a secreted protein. We found WIF1 in ~4% of total nuclei of the infarct border zone at MI-day 3, with a fraction

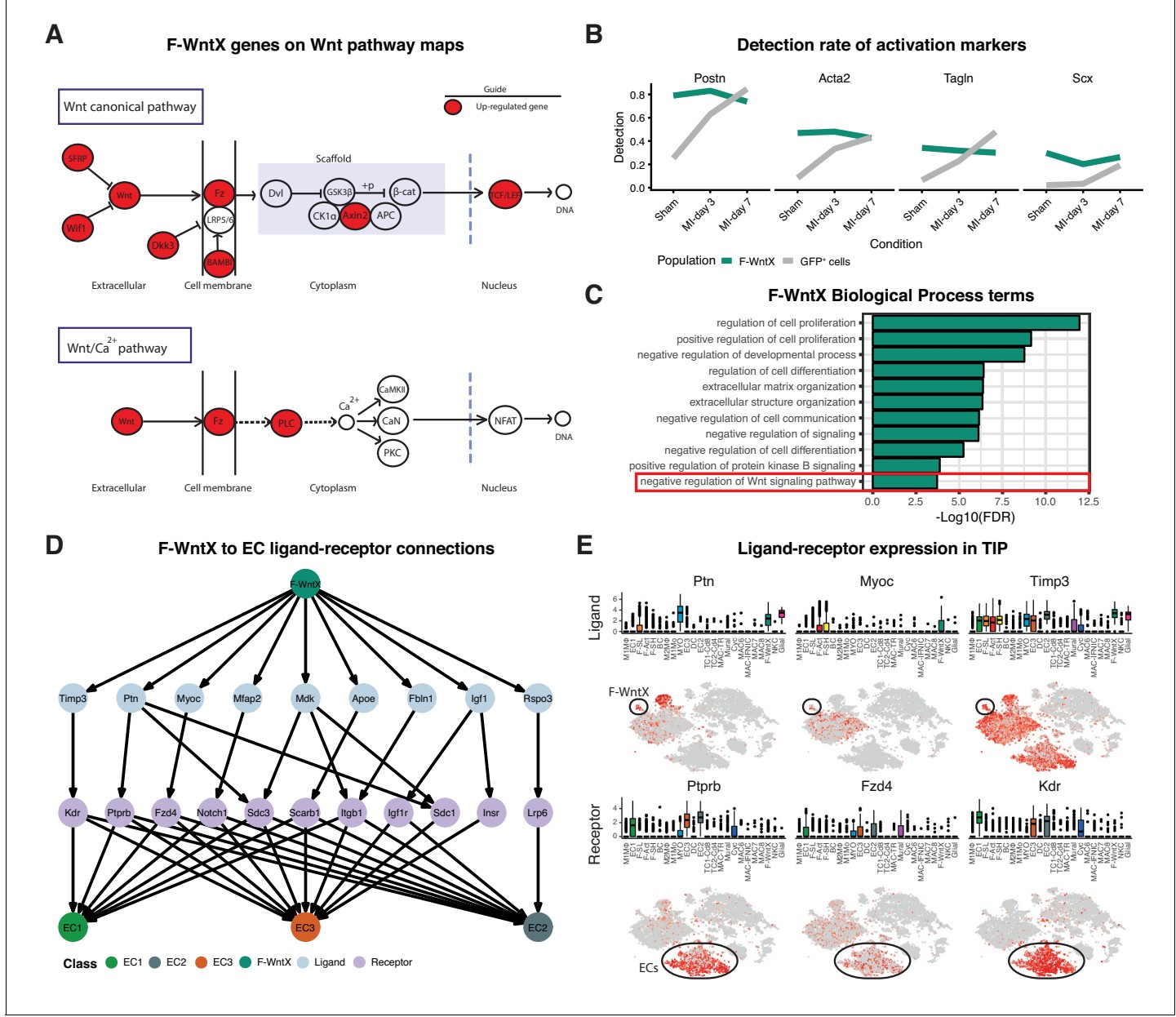

**Figure 5.** Features of the F-WntX population. (**A**) Differentially expressed genes in F-WntX overlaid on Wnt pathway maps. Wnt node includes genes *Wnt5a* and *Wnt16*. (**B**) Detection rate, representing the percentage of cells expressing a gene, across all conditions for cells in F-WntX or all GFP+ cells combined. (**C**) Example GO BP terms over-represented (FDR < 0.05) in genes upregulated in F-WntX compared to F-SL/F-SH populations. (**D**) Tree plot showing ligand-receptor connections from F-WntX to EC sub-populations as calculated in TIP. Top node refers to source population, second layer to ligands, third layer to receptors and leaf nodes represent target populations. (**E**) Examples of F-WntX:EC ligand and corresponding receptor expression as visualised in box and t-SNE plots. For each ligand the corresponding receptor is immediately below.

DOI: https://doi.org/10.7554/eLife.43882.024

The following figure supplement is available for figure 5:

**Figure supplement 1.** Activation and paracrine signature of F-WntX cells.

DOI: https://doi.org/10.7554/eLife.43882.025

of these (~5%) being GFP+ by IF, indicating a fibroblast identity (*Figure 6C,I*), and overall ~17% were positive for Ki67 (*Figure 6E,I*). WIF1+ cells were negative for CD31, and negative or very low for αSMA, with rare exceptions (*Figure 6F,F',H*). However, WIF1 was also expressed in ~4% of total CD45+ cells in the infarct border zone (~15% of WIF1+ cells were CD45+) (*Figure 6G,I*). We observed

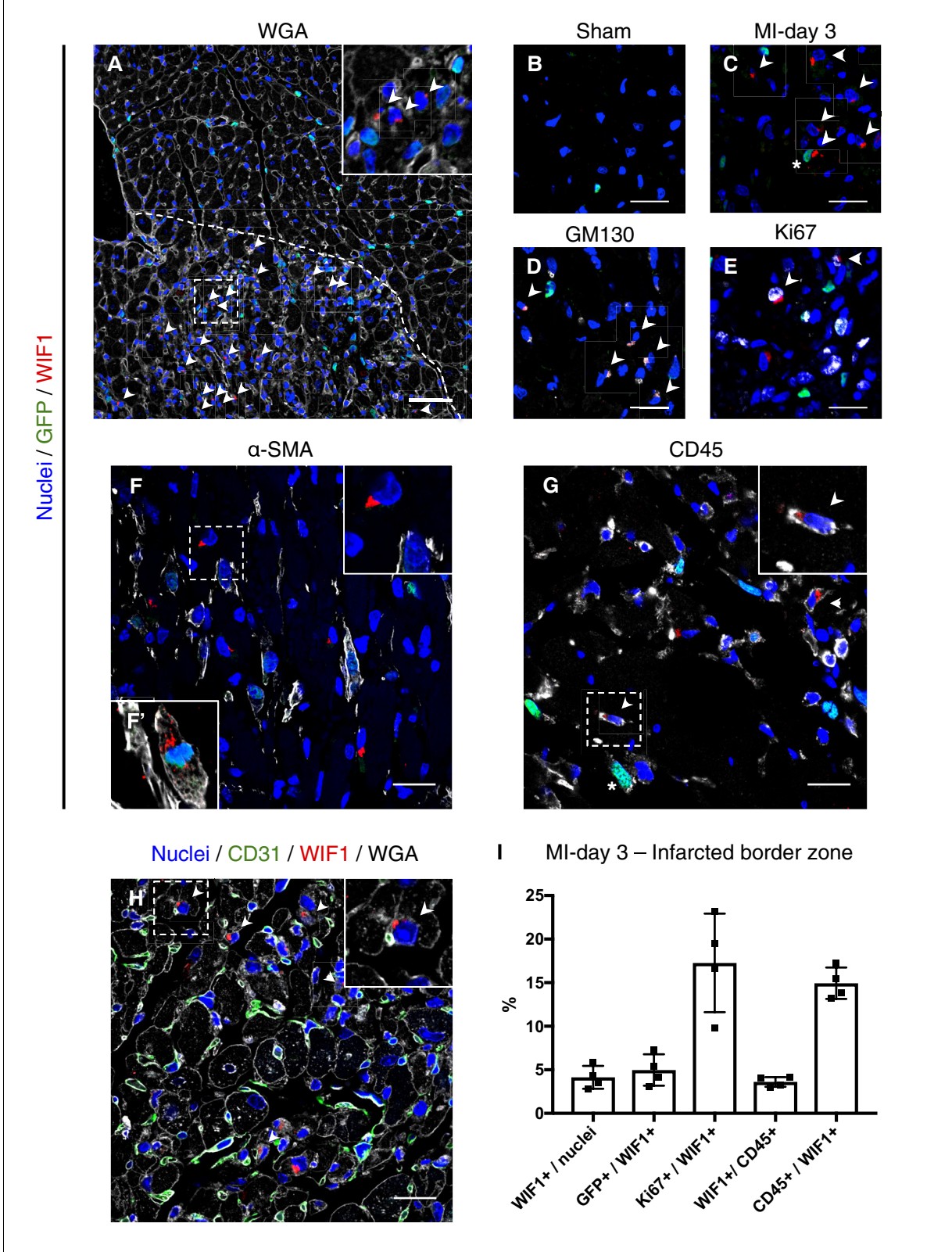

**Figure 6.** WIF1 localization and co-expression in injured and uninjured hearts of *Pdgfra*-GFP+ mice. (**A**) Representative image of WIF1 (red), GFP (green) and Wheat Germ Agglutinin (WGA, grey) co-immunostaining showing the border zone at MI-day 3. Arrowheads show WIF1+ cells. Scale bar - 50 μm. (**B–C**) Representative images of WIF1 (red) and GFP (green) co-immunostainings showing left ventricle (sham, (**B**) or infarcted border zone at MI-day 3 (**C**). Arrowheads show WIF1+ cells, Asterix shows WIF1+GFP+ cells. Scale bars - 20 μm. (**D–G**) Representative images of co-immunostainings for WIF1

*Figure 6 continued on next page*

**Figure 6 continued**

(red), GFP (green) and markers (gray) for golgi (GM130, D, Arrowheads show WIF1⁺GM130⁺ cells), proliferation (Ki67, E, Arrowheads show WIF1⁺Ki67⁺ cells), smooth muscle cells and myofibroblasts (α-SMA, F, (F') showing an example of a WIF1⁺GFP⁺α-SMA⁺ cell from another section), and leukocytes (CD45, G, Arrowheads show WIF1⁺CD45⁺ cells, Asterix shows GFP⁺CD45⁺ cells). Scale bar - 20 μm. (H) Representative image of co-immunostaining for WIF1 (green), WGA (gray) and endothelial cell marker CD31 (green). Arrowheads show WIF1⁺ cells in close proximity/contact with CD31⁺ cells. Scale bar - 20 μm. (I) Quantification of marker-positive cells in the infarcted border zone of MI-day 3 hearts. n = 4.
DOI: https://doi.org/10.7554/eLife.43882.026

The following source data and figure supplements are available for figure 6:

**Source data 1.** Source data for quantification of marker-positive cells summarized in *Figure 6I*.
DOI: https://doi.org/10.7554/eLife.43882.029
**Figure supplement 1.** WIF1 expression pattern in E14.5 embryos.
DOI: https://doi.org/10.7554/eLife.43882.027
**Figure supplement 2.** WIF1 protein expression after sham or MI.
DOI: https://doi.org/10.7554/eLife.43882.028

frequent close proximity or contact between WIF1⁺ cells and CD31⁺ ECs in tissue sections (*Figure 6H*), in line with the predicted cell-cell ligand-receptor connection between F-WntX cells and ECs (*Figure 3A*). Such proximity was less obvious for α-SMA⁺ or CD45⁺ cells (*Figure 6F,G*). Overall, our data suggest that WIF1 expression is post-transcriptionally regulated and injury-dependent, appearing in the infarct border-zone at MI-day 3 in a subset of fibroblasts and immune cells. The temporal window of WIF1 expression overlaps with fibroblast activation and expansion, and the beginning of EC renewal and MYO differentiation.

## Flux of fibroblasts and myofibroblasts after MI

MI is associated with appearance of activated fibroblasts and myofibroblasts. MI-day 7 was characterized by the appearance of a large population of myofibroblasts (MYO), representing 52% of GFP⁺ cells at MI-day 7 in our data (*Figure 4A–D*). MYO showed strong upregulation of numerous collagen genes (e.g. *Col1a1*, *Col3a1*, *Col5a2*), as well as *Postn* (99.5%) and *Acta2* (61%) at high levels, indicative of an activated state and suggestive of a contractile phenotype for a subset of cells (*Tallquist and Molkentin, 2017*; *Travers et al., 2016*) (*Figure 4E*; *Figure 4—figure supplement 3*). Upregulated genes involved in wound healing and cell migration included *Fn*(*Fibronectin*) and *Cthrc1*(*Collagen Triple Helix Repeat Containing I*), the latter representing a highly specific marker for MYO (*Figure 4F*). MYO showed decreased expression of *Pdgfra*, *Pdgfra*-GFP, *Ly6a*(*Sca1*), *Thy1* (*Cd90*) and *Cd34* (*Figure 4E*; *Figure 4—figure supplement 2C*; *Supplementary file 4*), indicating loss of stem/progenitor cell markers.

Earlier in the injury process, there was a distinct increase in a population with a signature consistent with activated fibroblasts (Fibroblast: activated; F-Act). These represented 48% of GFP⁺ cells at MI-day 3, before diminishing to 12% at MI-day 7 (*Figure 4A–D*). F-Act expressed *Postn* at high levels in ~80% of cells (*Figure 4E*) consistent with an activated state (*Tallquist and Molkentin, 2017*). They expressed *Acta2* in 28% and 35% of cells at MI-day 3 and MI-day 7, respectively, although at much lower levels compared to MYO, suggesting an emerging contractile phenotype in some cells. On the population dendrogram, F-Act was most closely related to fibroblast populations (F-SH, F-SL, F-Trans) and was more distant from MYO (*Figure 4C*). Whereas F-Act expressed few genes that could be considered highly specific, the top upregulated gene was *Cilp* (*Figure 4F*), encoding a matricellular protein and inhibitor of TGF-β1 signaling, consistent with F-Act being a pre-MYO population in which fibrosis is constrained. The expansion of F-Act at MI-day 3 correlated with a decrease in the proportion of F-SH and F-SL, whereas the diminishment of the F-Act population at MI-day 7 coincided with the appearance of MYO and an apparent partial restoration of F-SH and F-SL cells (*Figure 1A,E*; *Figure 4A,D*; see Discussion).

A distinct GFP⁺ population contained fibroblasts undergoing proliferation (Fibroblast - cycling; F-Cyc), comprising 15% of GFP⁺ cells at MI-day 3 and 3% at MI-day 7 (*Figure 4D*), consistent with studies showing peak fibroblast proliferation at MI-days 2–4 (*Fu et al., 2018*; *Ivey et al., 2018*). F-Cyc uniquely expressed a strong cell cycle gene signature, including *Ccnb2*(*CyclinB*), *Cdk1*(*Cyclin dependent kinase 1*) and *Mki67*(*Ki67*) (*Figure 4E*; *Figure 4—figure supplement 3*), and expressed both *Postn* (88%) and *Acta2* (76%) at high levels (*Figure 4E*; see below).

## Cell trajectory analysis of *Pdgfra*-GFP⁺ cells

To look at potential relationships between the major GFP⁺ populations, we analyzed cell trajectories using *Diffusion Maps* (Materials and methods). MYO, F-WntX and F-Cyc were represented as three different trajectories along diffusion components 1, 2 and 3, respectively (*Figure 7A*), with the root containing the two large unactivated fibroblast populations F-SH and F-SL, which were most prominent in sham hearts. F-Trans was an intermediary population along the trajectory to F-WntX, and F-Act was an intermediary population for both F-Cyc and MYO branches. F-Cyc, characterized by expression of a strong cell cycle gene signature, was represented most strongly at MI-day 3, whereas MYO was exclusively associated with MI-day 7 (*Figure 7B,C*). These data suggest that F-Act expands by proliferation up to MI-day 3 (F-Cyc trajectory) and differentiates to MYO during the transition from MI-day 3 to MI-day 7. The presence of some F-Cyc-like cells between the F-Cyc and MYO trajectories at MI-day 7 raises the possibility that a small fraction of F-Act cells differentiate rapidly into MYO after or during division (see below).

We examined DE and GO BP terms in F-Act, F-Cyc and MYO compared with F-SL and F-SH combined, across all conditions (*Figure 7D*; *Supplementary file 5*; *Supplementary file 6*). DE genes for F-Act were over-represented in terms for *collagen fibril organization* (including several collagen genes) and *regulation of wound healing*. Many of these genes were also expressed in F-Cyc and MYO; however, there was an additional large gene signature strongly upregulated in MYO compared to F-Act (*Figure 7D*). DE genes for MYO demonstrated GO BP terms for *collagen fibril organization* and *cell adhesion*, containing collagen genes *Col3a1*, *Col5a1*, *Col11a1* and *Col14a1*, and others involved in cell:cell and cell:matrix adhesion including *Thbs1* (encoding Thrombospondin 1) and *Fbn1*. Other terms included *angiogenesis* and *heart development* as well as *negative regulation of canonical Wnt signaling pathway*, containing many genes previously identified in F-WntX.

## Minor *Pdgfra*-GFP⁺ populations

Minor GFP⁺ populations included epicardial cells (EPI), observed only at MI-day 7. These expressed *Wt1* (*Figure 4—figure supplement 3*) and overall were related transcriptionally to dissected adult mouse epicardium (*Bochmann et al., 2010*) (Spearman's correlation test, p=0.014, *r* = 0.26). These are likely to be epicardial-derived fibroblasts that arise after MI as the epicardium reactivates its developmental program (including *Pdgfra* expression) (*Zhou et al., 2011*). Consistent with a previous cardiac scRNA-seq analysis on uninjured hearts (*Skelly et al., 2018*), we did not detect epicardial cells in TIP data, suggesting that these cells are under-sampled in our experiments - this is likely technical as epicardial cells have been detected readily in single-nucleus RNA-seq (*Hu et al., 2018*).

The minor population F-IFNS (Fibroblast: Interferon stimulated), found in all conditions, was negative for *Cd45* and expressed high levels of *Col1a1* and other fibroblast markers (*Figure 4E*; *Figure 4—figure supplement 3*), demonstrating a fibroblast identity, and interferon-responsive genes (*Figure 4F*; *Supplementary file 4*) (*Zhou et al., 2013*). Other minor GFP⁺ populations were EC and MAC (*Figure 4A–C*), which had EC and MΦ identities, respectively (Materials and methods).

## Transcription factors expressed in *Pdgfra*-GFP⁺ cells

Several TF genes expressed in GFP⁺ cells may drive differentiation or responses to environmental stimuli (*Figure 4—figure supplement 4*). *Scleraxis*, already mentioned, was expressed in F-WntX cells and MYO. The basic helix-loop-helix factor gene, *Tcf21*, an accepted marker of cardiac fibroblasts (*Tallquist and Molkentin, 2017*), was expressed in most GFP⁺ populations with the exception of F-WntX cells. T-box factor gene *Tbx20*, another fibroblast marker, was expressed across all GFP⁺ cells and upregulated in F-WntX. The homeodomain TF gene, *Meox1*, which is part of the cardiac fetal gene expression signature reactivated in injured hearts (*Lu et al., 2018*), showed upregulation in subsets of GFP⁺ cells, most prominently in activated populations (F-Act, F-Wntx, F-Cyc and MYO), and may drive the activated state. *Csrp2*, *Zfp385a* and *Hmgb2* expression was also restricted among populations, with *Csrp2* and *Zfp385a* showing strikingly complementary patterns. Interestingly, the homeodomain TF gene, *Prrx1*, which in BM is expressed in a subset of mesenchymal cells with CFU-F and multi-lineage differentiation potential (*Kfoury and Scadden, 2015*), was expressed across all GFP⁺ populations in all conditions.

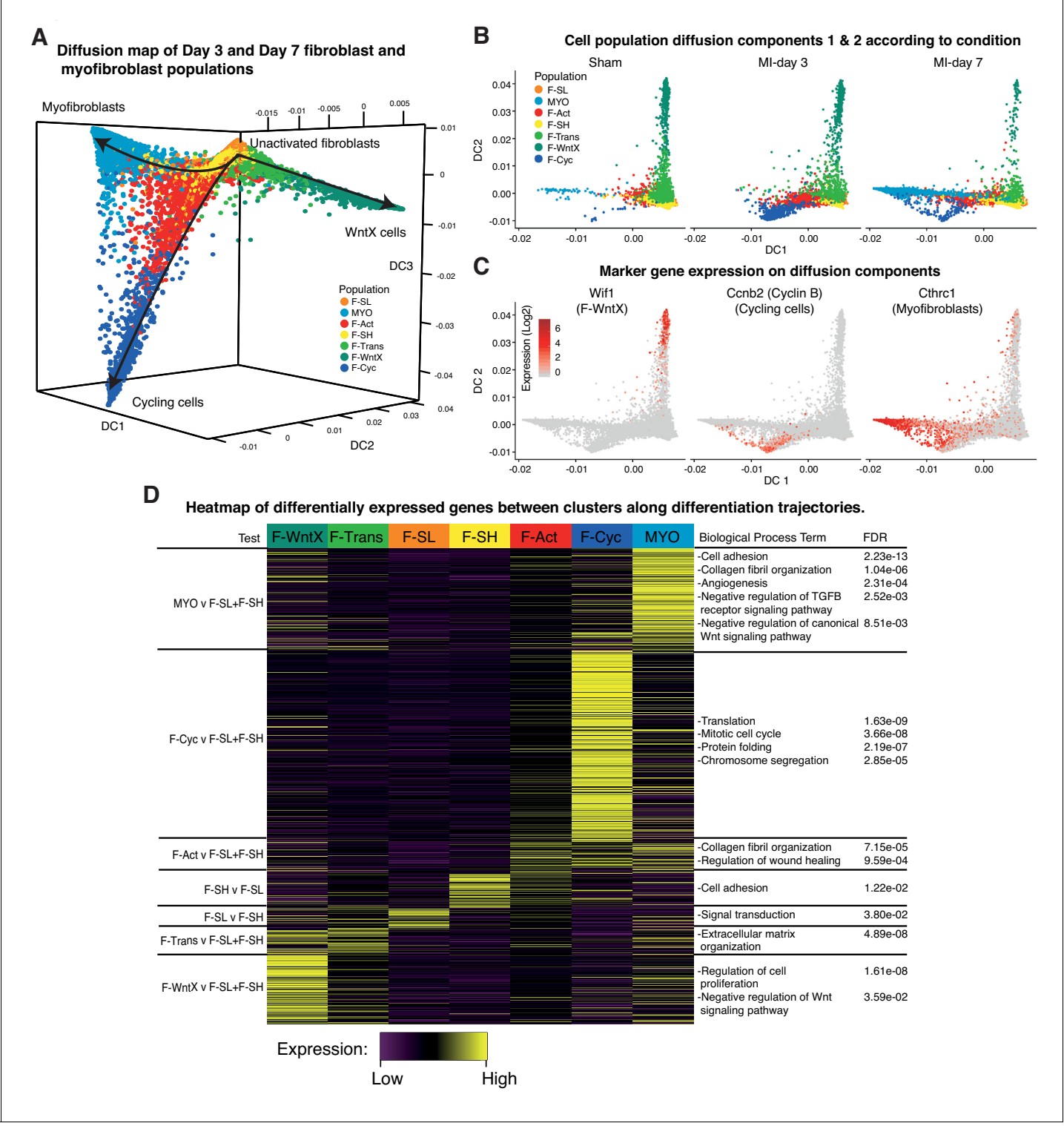

**Figure 7.** Diffusion Map analysis of GFP[+] cells. (**A**) 3D Diffusion Map of main fibroblast/myofibroblast populations with cells colored according to population. (**B**) 2D Diffusion Map facetted according to experimental condition. (**C**) Expression of marker genes on main trajectories of diffusion components across conditions. (**D**) Heatmap of differentially expressed genes with representative GO Biological Process terms.

DOI: https://doi.org/10.7554/eLife.43882.030

## Activated fibroblast and myofibroblast sub-populations

We re-clustered the sham and MI datasets at days 3 and 7 individually (*Figure 8A,B*), and repeated *Diffusion Map* analysis of GFP⁺ fibroblast lineages (*Figure 8C,D*) (Materials and methods). Whereas most populations identified at day 3 directly corresponded to those identified in the aggregate analysis, we found that F-Cyc could now be sub-divided into two populations, with only one exhibiting a clear cell cycle signature (*Figure 8E*, *Supplementary file 7*; *Supplementary file 8*). An intermediary population (Fibroblast - Cycling Intermediate; F-CI) showed upregulation of fibroblast activation markers including *Postn*, *Cthrc1* and *Acta2* (*Figure 8—figure supplement 1A*), but did not express markers of cell cycle, suggesting that it represents an additional population of activated fibroblasts, potentially competent for cell cycle entry. F-CI also upregulated genes involved in protein translation, a signature absent in F-Act (*Figure 8E*). The translation signature was maintained, albeit in attenuated form, in F-Cyc. Based on an iRF classifier trained to predict MI-day 3 populations (*Figure 8—figure supplement 1B*), we found no corresponding F-CI cluster at MI-day 7, indicating its transient nature (*Figure 8—figure supplement 1C,D*). *Diffusion Map* analysis also lent weight to the hypothesis that F-CI is a transitory population between F-Act and F-Cyc at MI-Day 3 (*Figure 8C*).

We next removed genes annotated with the GO term 'Cell Cycle' and re-clustered populations. At MI-day 3, 88% of F-Cyc cells merged with F-CI, strongly supporting the hypothesis that F-CI is a pre-proliferative precursor of F-Cyc (*Figure 8—figure supplement 1E*). After removal of cell cycle genes at MI-day 7, 65% of F-Cyc cells remained as a distinct (proliferative) population, although 26% merged with F-Act and 6% merged with the MYO populations (*Figure 8—figure supplement 1F*). Taken together with *Diffusion Map* analysis, these data indicate that the closest population to the majority F-CI and cycling cells is F-Act rather than MYO, even though both express *Acta2* and other MYO markers at high levels (*Figure 4E*). However, a minority of F-Cyc cells at MI-day 7 may be dividing MYO cells or in transition to MYO (see Discussion). Our data support the idea that F-CI cells are an activated form of fibroblast closely related to F-Cyc and derivative of F-Act. We hypothesize that they are primed for cell cycle entry and differentiation, but this requires further investigation.

In the MI-day 7 re-analysis, we found that MYO could be sub-divided into three clusters - MYO-1, MYO-2 and MYO-3 (*Figure 8B,F*; *Supplementary file 9*; *Supplementary file 10*), and comparing these clusters we noted genes corresponding to the contrasting functions of fibrosis inhibition and promotion. MYO-2 upregulated *Tgfb1* (encoding TGF-β1), which is one of the strongest and most studied drivers of MYO formation (*Figure 8G*; *Supplementary file 9*), *Scx*(*Scleraxis*), which regulates ECM production and the myofibroblast phenotype downstream of TGF-β (*Bagchi et al., 2016*) and *Thbs4*(*Thrombospondin 4*), a regulator of cardiac fibrosis (*Frolova et al., 2012*). We sourced RNA-seq data from cultured mouse cardiac fibroblasts untreated or treated with TGF-β (*Schafer et al., 2017*) and extracted DE genes. As expected, the highest positive correlations with TGF-β treatment (log² fold-changes) were in MYO and other activated fibroblast populations (*Supplementary file 11*). For inter-MYO comparisons, we found a significant positive correlation with TGF-β treatment in DE genes comparing MYO-2 v MYO-1 (Spearman's correlation test, $r$ = 0.26, $P_{adj}$ = 1.93e-12) and MYO-3 v MYO-1 (Spearman's correlation test, $r$ = 0.25, $P_{adj}$ = 1.75e-10), supporting the conclusion that MYO-2 and MYO-3 are pro-fibrotic.

In contrast, MYO-1 showed strong upregulation of anti-fibrosis genes included *Wisp2*, encoding matricellular protein CCN5 which can reverse established fibrosis (*Jeong et al., 2016*), and *Sfrp2* encoding a soluble WNT receptor and antagonist of canonical WNT signalling, which is pro-fibrotic (*Mirotsou et al., 2007*) (*Figure 8G*). MYO-1 upregulated genes showed significant GO terms *negative regulation of growth factor stimulus* and *negative regulation of cell proliferation* (*Figure 8F*; *Supplementary file 10*), which included *Sfrp1*, implicated in inhibition of fibroblast proliferation and fibrosis (*Sklepkiewicz et al., 2015*), and *Htra1* and *Htra3*, implicated in TGF-β signaling inhibition (*Tocharus et al., 2004*). *Diffusion Map* analysis of MYO sub-populations showed that MYO-1 and MYO-2 were distinct clusters with some overlap, suggesting a continuum of states, whereas MYO-3 did not appear to be a distinct population (*Figure 8—figure supplement 1G*).

## Discussion

The mammalian heart is composed of a complex interdependent community of cells, although their interactions and flux are poorly characterized. Here, we present scRNA-seq data on >30,000 individual cardiac interstitial cells from sham, and MI days 3 and 7 hearts. Our interrogation of both the

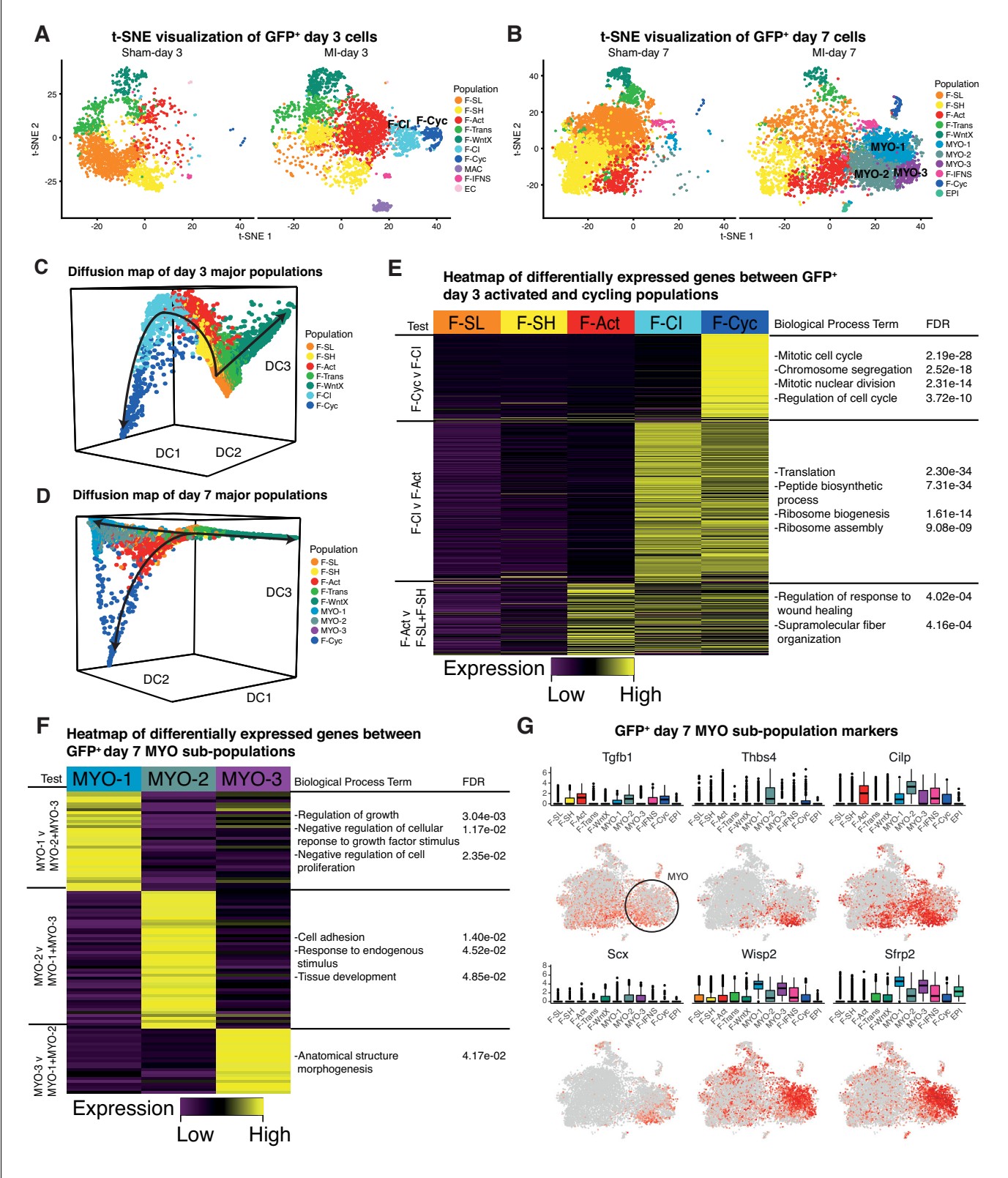

**Figure 8.** Time-point-specific analysis of GFP+ scRNA-seq. (**A,B**) t-SNE visualization of GFP+ populations 3 days post sham/MI (**A**) and 7 days post sham/MI (**B**). (**C,D**) 3D Diffusion Map analysis of day 3 major populations (**C**) and day 7 major populations (**D**). (**E**) Heatmap of upregulated genes in day 3 injury-response populations. (**F**) Heatmap of differentially expressed genes between myofibroblast sub-populations. (**G**) Gene expression visualized in box and t-SNE plots for myofibroblast sub-population marker genes.

*Figure 8 continued on next page*

*Figure 8 continued*

DOI: https://doi.org/10.7554/eLife.43882.031

The following figure supplement is available for figure 8:

**Figure supplement 1.** Reanalysis of GFP-day 3 and GFP-day 7 data-sets separately.

DOI: https://doi.org/10.7554/eLife.43882.032

total interstitial population (TIP) and flow-sorted *Pdgra*-GFP⁺CD31⁻ fibroblast lineage cells has given us a high-resolution map of cell lineage, state and flux in the healthy and injured heart, considerably extending preliminary studies (*Kanisicak et al., 2016*; *Skelly et al., 2018*; *Gladka et al., 2018*).

On the basis of these data, resident fibroblasts could be segregated into two major sub-populations denoted *Sca1*^high (F-SH) and *Sca1*^low (F-SL), both expressing canonical fibroblast markers such as *Pdgfra*, *Pdgfra*-GFP, *Ddr2* and *Col1a1*. F-SH cells were enriched in S⁺P⁺ (SCA1⁺PDGFRα⁺) fibroblasts and clonal colony-forming units (*Pinto et al., 2016*; *Chong et al., 2011*; *Noseda et al., 2015*), and F-SH and F-SL showed distinct adhesive and secretory phenotypes, highlighting the likely functional differences between them.

We describe a novel activated fibroblast population, F-WntX, in sham hearts, which persists after MI. The related F-Trans is an intermediary population between F-SL and F-WntX (*Figure 9*). Aside from a low proportion of activated fibroblasts (F-Act) present in sham hearts, no other sham population (>75% of GFP⁺ cells) showed an activated phenotype. When we re-analyzed the scRNA-seq data of Skelly *et al.* on interstitial cells from uninjured hearts (*Skelly et al., 2018*), we identified all of the main fibroblast populations that we describe here in sham hearts (*Figure 4—figure supplement 5A–C*). F-Act and F-WntX expressed activation marker *Postn* in both studies; however, in contrast to our study, all populations identified by *Skelly et al.*, including endothelial and immune cells, expressed the contractile marker *Acta2*. The reason for this is unclear, but is likely technical. Gladka *et al.* profiled cardiac populations after ischemia-reperfusion injury using scRNA-seq and highlighted *Ckap4* as a novel marker of activated fibroblasts (*Gladka et al., 2018*); however, our data shows

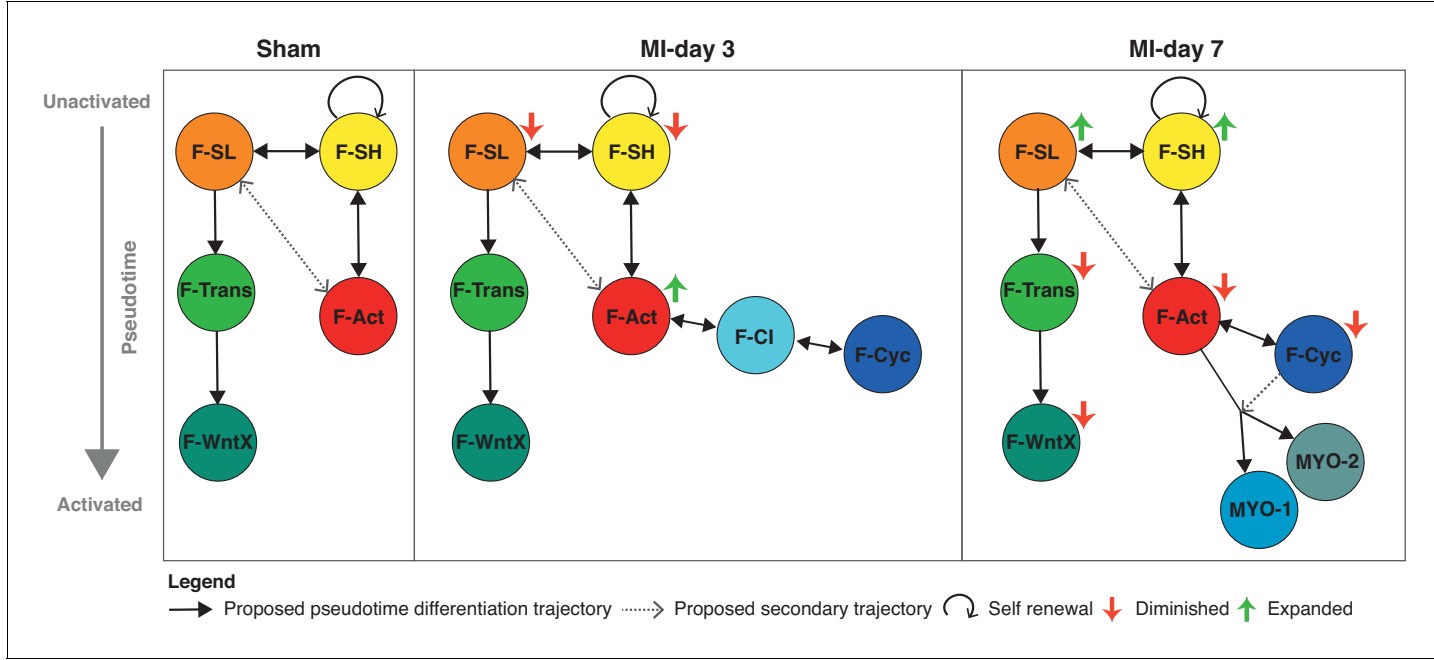

**Figure 9.** Schematic summary of the flux and pseudotime differentiation dynamics of GFP⁺ populations between sham, MI-day 3 and MI-day 7. Populations are ordered in pseudotime from unactivated (top) to most activated/mature (bottom). Arrows connecting populations indicate direction of proposed differentiation/pseudotime trajectory. Colored arrows indicate whether the population appears to expand or diminish relative to the previous time-point.

DOI: https://doi.org/10.7554/eLife.43882.033

*Ckap4* expression across virtually all cardiac stromal populations (*Figure 4—figure supplement 5D*), a discrepancy that may relate to the relatively low number of cells profiled in the Gladka *et al.* study.

Among fibroblasts, F-WntX uniquely expressed *Wif1*, encoding a canonical and non-canonical WNT signaling antagonist (*Meyer et al., 2017*; *Bányai et al., 2012*). WNT pathways play complex roles in cardiac biology and disease, impacting immune, vascular and pro-fibrotic pathways, and many drugs inhibiting WNT signaling are under investigation for their impacts on heart repair (*Foulquier et al., 2018*; *Palevski et al., 2017*). WIF1 additionally inhibits signaling through CTGF, a polyfunctional matricellular protein and positive driver of fibrosis (*Travers et al., 2016*; *Surmann-Schmitt et al., 2012*). WIF1 is a tumor suppressor inhibiting tumour angiogenesis through both WNT and VEGF pathways (*Ko et al., 2014*; *Hu et al., 2009*). *Wif1* knockout mice show inhibition of Mo differentiation and abnormal chamber remodeling after MI (*Meyer et al., 2017*), while un-regulated transgenic WIF1 expression causes dilated cardiomyopathy (*Lu et al., 2013*), collectively indicating that correctly regulated WIF1 positively contributes to cardiac repair. One source of WNT proteins is inflammatory macrophages, and myeloid-specific deletion of the essential WNT transporter *Wntless* leads to improved cardiac functional recovery after MI involving an increase in reparative M2-like macrophages and angiogenesis (*Palevski et al., 2017*). In addition to WIF1, F-WntX cells showed upregulation of other WNT and TGFβ pathway antagonists (*Figure 5A*), overall flagging F-WntX cells as paracrine mediators of an anti-WNT/CTGF/TGFβ signaling milieu essential for cardiac repair.

WIF1 protein expression occurred in the border zone at MI-day 3, but not in sham or MI-days 1 and 7 hearts, consistent with previous findings (*Meyer et al., 2017*). We also detected WIF1 in ~4% of CD45$^+$ immune cells infiltrating the border zone. We acknowledge that our IF studies may have underestimated the number of WIF1$^+$ cells; for example, if we were unable to detect cells actively secreting WIF1 but lacking protein accumulation in the golgi. Contrary to our results, *Meyer et al.* showed WIF1 levels peaking at MI-day 1 and diminished by MI-day 7 using western blotting (*Meyer et al., 2017*). Whereas these differences need resolving, *en face* our data suggest that WIF1 expression is post-transcriptionally regulated with a peak around MI-day 3, consistent with our proposed function for WIF1 and the F-WntX population generally in inhibiting fibrosis and angiogenesis, and promoting differentiation of Mo, during the transition between the inflammatory and fibrotic phases of heart repair.

At MI-day 3, both F-SH and F-SL were significantly diminished, and we hypothesize that this occurs as they convert to an activated state (F-Act; *Postn*$^+$*Acta2*$^{negative-low}$). The scale of conversion suggests that fibroblasts remote from the infarct also become activated. Whereas F-Act upregulated genes that were associated with *collagen fibrils* and *wound healing* in common with MYO, F-Act cells are by far more closely related to resident fibroblasts than to MYO. A proportion of F-Act cells were actively proliferating at MI-day 3 and to a lesser extent at MI-day 7, consistent with the known peak of fibroblast proliferataion (*Fu et al., 2018*; *Ivey et al., 2018*). At MI-day 3, we also identified an activated, non-proliferating fibroblast population (F-Cl) situated between F-Act and F-Cyc in trajectory plots, showing strong upregulation of genes supporting protein translation, and we hypothesize that this secondary activated state occurs in readiness for cell division and differentiation.

MYO was evident only at MI-day 7, where they represented >50% of total GFP$^+$ cells. This may be an underestimate, as the Chromium microfluidic device biases against larger cells, and some MYO cells may be GFP$^-$. Highlighting the limitations of using the contractile marker αSMA to define MYO (*Tallquist and Molkentin, 2017*), a proportion of F-WntX, F-Act, F-Cyc and MYO cells expressed its gene, *Acta2*. MYO cells massively upregulated a distinct network of genes related to *cell adhesion*, *collagen fibril organization* and *angiogenesis*, consistent with their known roles (*Shannon et al., 2003*; *Tallquist and Molkentin, 2017*). Top ECM genes in these categories such as *Postn*, *Fn1* and *Col8a1* were expressed in virtually all MYO cells. However, our stage specific analysis of GFP$^+$ cells at MI-day 7 allowed us to discern three distinct MYO sub-populations expressing pro-fibrotic (MYO-2 and MYO-3) or anti-fibrotic (MYO-1) states. MYO-1 expressed anti-TGF-β, anti-WNT and anti-proliferative signatures. Consistent with the view that MYO differentiation involves multiple cellular states, *Fu et al. (2018)* recently described a population of fibrobast-derived 'matrifibrocytes', quiescent cells which persist in the mature scar after MI.

The three major trajectories predicted by our *Diffusion Map* pseudotime analysis offer new insights into cardiac homeostasis and repair. The directionality of trajectories is suggested by the fact that all appear rooted in the resident F-SH and F-SL fibroblasts. One major trajectory arises

specifically from F-SL and transits through F-Trans to F-WntX as the terminal state (*Figure 9*). Neither F-Trans nor F-WntX proliferate and may be primed for involvement in an injury response without the need for expansion. Up to MI-day 3, F-Act cells proliferate, but do not differentiate to MYO, showing that these events can be uncoupled. Differentiation to MYO occurs after MI-day 3, and our trajectory data suggest that this is largely from non-proliferating F-Act cells. We found no evidence for significant proliferation of MYO, although earlier time points need to be analyzed.

Resident fibroblasts F-SH and F-SL were depleted at MI-day 3, likely as a result of their activation and proliferation, and showed an apparent restoration by MI-day 7 in both TIP and GFP⁺ cells (falling just short of our stringent p-value of 0.01 for DPA) (*Figure 1A,E*; *Figure 4A,D*; *Figure 8A,B*). A caveat of pseudotime trajectories is that they may be bi-directional. If confirmed, the restoration of F-SH and F-SL between MI-days 3–7 (after the main fibroblast proliferative period [*Fu et al., 2018*; *Ivey et al., 2018*]) points to renewal involving phenotypic regression. Regression of myofibroblasts to a less activated state was proposed recently (*Kanisicak et al., 2016*), although our data suggest that the *Postn*-Cre lineage tracing mice used in that study to mark myofibroblasts labels most if not all F-Act cells (*Figure 4E*). Certainly, the mechanism of activation, proliferation, self-renewal, differentiation and de-differentiation of cardiac fibroblasts warrants deeper investigation.

The Mo/MΦ lineages of the heart have diverse functions, including in immunity, removal of debris and protection against autoimmunity during the early phases of MI, while promoting repair in latter phases, remodeling of the fetal coronary tree, neonatal heart regeneration and atrioventricular conduction (*Lavine et al., 2014*; *Hulsmans et al., 2017*; *Leid et al., 2016*; *Aurora et al., 2014*; *Nahrendorf et al., 2007*). The adult heart, like other organs, contains resident MΦ, some of which have their origins in erythromyeloid progenitors in the embryonic yolk sac and fetal liver (*Leid et al., 2016*) and which self-renew during homeostasis and injury (*Heidt et al., 2014*; *Epelman et al., 2014*; *Dick et al., 2019*). Other resident populations have differing degrees of monocyte dependence and some may eventually be supplanted during injury and aging by blood-born monocytes (*Molawi et al., 2014*; *Dick et al., 2019*). These resident MΦ likely have roles in defense against infection, antigen presentation and stimulation of T-cell responses, as well as efferocytosis (*Epelman et al., 2014*). In the neonatal heart, they are though to be essential for regeneration through stimulation of CM proliferation and angiogenesis (*Lavine et al., 2014*), and in the adult may be important for limiting fibrosis (*Lavine et al., 2014*; *Dick et al., 2019*), functions similar to pro-reparative M2 MΦ.

Our scRNA-seq data identified most of the known Mo/MΦ populations highlighted by targeted scRNA-seq of cardiac Mo/MΦ cells (*Dick et al., 2019*), including *Ccr2*⁺ and *Ccr2*⁻ tissue resident MΦ, pro-inflammatory M1 Mo and MΦ at MI-day 3, and non-classical M2 MΦ at MI-day 7, as well as several minor populations and inflammatory fibroblasts. These data, combined with communication maps, provide a preliminary framework for further analysis of their relationships and flux in homeostasis, different disease models and after therapeutic interventions. Our *Diffusion Maps* lineage trajectory shows a continuum of states from M1Mo through M1MΦ to M2MΦ across the injury response, consistent with the recognised plasticity of blood born Mo (*Lavine et al., 2018*; *Nahrendorf and Swirski, 2016*). However, whether these states are determined exclusively within the injury environment remains to be determined. Our trajectory also shows convergence of M2MΦ with tissue-resident MΦ (*Figure 2—figure supplement 1C*). An emerging theme in the field is the similarity between yolk sac-derived tissue-resident MΦ and subsets of blood-born MΦ that appear during the reparative phase of MI (*Dick et al., 2019*). However, the latter may not fully adopt the gene expression signature of resident cells, nor fully compensate for their proposed functions, as suggested in the context of genetic ablation of *Cc3cr1*⁺ tissue-resident MΦ, although this may relate to the timing of their deployment (*Dick et al., 2019*). Irrespective of whether such cells are identical, it is noteworthy that pro-repairative macrophages appear to have multiple developmental origins, as found for virtually all other major adult heart lineages including CMs, ECs, SMCs, fibroblasts and adipocytes.

Our scRNA-seq data can be mined for expression of genes implicated in other forms of heart disease. For example, of 167 genes proximal to single nucleotide polymorphisms implicated in GWAS studies in atrial fibrillation (AF) (*Roselli et al., 2018*), 119 (71%) showed expression in our TIP single-cell data (for examples see *Figure 1—figure supplement 8*), suggesting possible involvement of stromal cells in AF risk.

We have identified substantial non-linear dynamics in the interactive cell communities of the heart (*Figure 9*), which, in this new light, can be further analyzed with lineage and functional tools. There remains a compelling clinical and economic rationale for finding new therapies for controlling the inflammation and fibrosis that accompanies virtually all forms of adult heart disease (*Gourdie et al., 2016*). In the long term, scRNA-seq analysis of cardiac homeostasis and disease will provide new entry points for discovering novel drugs and interventions supporting heart repair and regeneration.

# Materials and methods

## Key resources table

| Reagent type (species) or resource | Designation | Source or reference | Identifiers | Additional information |
|---|---|---|---|---|
| Gene (*Mus musculus*) | Pdgfra | NCBI | NCBI Gene ID: 18595, MGI:97530 | |
| Strain, strain background (*Musmusculus*, C57BL/6J) | Wild type, WT | The Jackson Laboratory, Stock Number: 000664, | RRID:IMSR:JAX:000664 | |
| Strain, strain background (*Musmusculus*, C57BL/6J) | *Pdgfra^tm11(EGFP)Sor^*; PDGFRa^GFP/+^ | The Jackson Laboratory, Stock Number: 007669, PMID: 12748302 | MGI:2663656 | |
| Antibody | APC-conjugated Rat monoclonal anti-mouse PDGFRa (CD140a) | eBioscience | 17-1401-81, Clone APA5 | (1:200) |
| Antibody | PE-Cy7-conjugated Rat monoclonal anti-mouse CD31 (PECAM-1) | eBioscience | 25-0311-82, Clone 390 | (1:400) |
| Antibody | PE-conjugated Rat monoclonal anti-mouse Sca1 (Ly6A/E) | BD Pharmingen | 553108, Clone D7 | (1:400) |
| Antibody | APC-Cy7-conjugated Rat monoclonal anti-mouse CD45 | BD Pharmingen | 557659, Clone 30-F11 | (1:400) |
| Antibody | PE-conjugated Rat monoclonal anti-mouse F4/80 | eBioscience | 12-4801-82, Clone BM8 | (1:400) |
| Antibody | APC-conjugated Rat monoclonal anti-mouse CD206 (MMR) Antibody | BioLegend | 141708, Clone C068C2 | (1:200) |
| Antibody | Chicken polyclonal anti-GFP | Abcam | ab13970 | (1:200) |
| Antibody | Rabbit polyclonal anti-Wif1 | Abcam | ab186845 | (1:1000) |
| Antibody | Rat monoclonal anti-CD31 | Dianova | DIA-310, Clone SZ31 | (1:100) |
| Antibody | Rat monoclonal anti-CD45 | BD Biosciences | 553076, Clone 30-F11 | (1:100) |
| Antibody | Mouse monoclonal anti-aSMA | Sigma | A2547, Clone 1A4 | (1:100) |
| Antibody | Rat monoclonal anti-Ki67 | Dako | M7249, Clone TEC-3 | (1:100) |
| Antibody | Mouse monoclonal anti-GM130 | BD Biosciences | 610822, Clone 35/GM130 | (1:400) |
| Antibody | Goat polyclonal anti-Chicken Alexa 488 | Life Technologies | A11039 | (1:500) |
| Antibody | Goat polyclonal anti-Rabbit Alexa 555 | Life Technologies | A21429 | (1:500) |

*Continued on next page*

*Continued*

| Reagent type (species) or resource | Designation | Source or reference | Identifiers | Additional information |
|---|---|---|---|---|
| Antibody | Goat polyclonal anti-Rabbit Alexa 680 | Life Technologies | A21109 | (1:500) |
| Antibody | Goat polyclonal anti-Rat Alexa 555 | Life Technologies | A21434 | (1:500) |
| Antibody | Donkey polyclonal anti-Mouse Alexa 594 | Life Technologies | A21203 | (1:500) |
| Antibody | APC-conjugated Rat monoclonal anti-mouse PDGFRa (CD140a) | eBioscience | 17-1401-81, Clone APA5 | (1:200) |
| Commercial assay or kit | Chromium Single Cell 30 Library and Gel Bead Kit v2 | 10x Genomics | 120237 | |
| Commercial assay or kit | Chromium Single Cell A Chip Kit | 10x Genomics | 120236 | |
| Commercial assay or kit | Chromium i7 Multiplex Kit | 10x Genomics | 120262 | |
| Commercial assay or kit | Nextera XT DNA Sample Preparation Kit (96 Samples) | Illumina | FC-131–1096 | |
| Commercial assay or kit | Nextera XT Index Kit v2 | Illumina | FC-131–2001 | |
| Commercial assay or kit | Fluidigm Single-Cell Auto Prep IFC chip (5–10 um) | Fluidigm | 100–5759 | |
| Commercial assay or kit | SMART-Seq v4 Ultra Low Input RNA Kit for the Fluidigm C1 System | Takara | 635026 | |
| Commercial assay or kit | NextSeq 500/550 High Output Kit v2 | Illumina | FC-404–2002 | |
| Other | LIVE/DEAD Viability/Cytoxicity Kit, for mammalian cells | Thermo Fisher Scientific | L-3224 | |
| Software, algorithm | CellRanger | 10x Genomics | https://support.10xgenomics.com/single-cell-gene-expression/software/downloads/latest | |
| Software, algorithm | STAR | PMID: 23104886 | https://github.com/alexdobin/STAR; RRID: SCR_015899 | |
| Software, algorithm | Bowtie 2 | PMID: 22388286 | http://bowtie-bio.sourceforge.net/bowtie2/index.shtml; RRID:SCR_005476 | |
| Software, algorithm | featureCounts | PMID: 24227677 | http://subread.sourceforge.net; RRID:SCR_012919 | |
| Software, algorithm | Seurat | PMID: 29608179 | https://satijalab.org/seurat/; RRID: SCR_007322 | |
| Software, algorithm | Destiny | PMID: 26668002 | https://bioconductor.org/packages/release/bioc/html/destiny.html | |
| Software, algorithm | PANTHER | PMID: 27899595 | http://www.pantherdb.org; RRID:SCR_004869 | |
| Software, algorithm | Iterative Random Forest | PMID: 29351989 | https://cran.r-project.org/web/packages/iRF/index.html | |
| Software, algorithm | Differential Proportion Analysis | This paper | *Source code 1* | Materials and methods: Differential proportion analysis |
| Software, algorithm | Cell communication analysis | This paper | *Source code 1* | Materials and methods: Ligand-receptor networks |

## Murine model

8-12 weeks old male mice were used in all experiments unless otherwise stated. For the single-cell RNA sequencing experiment, mice had a H2B-eGFP fusion gene knock-in at the endogenous *Pdgfra* locus (*Pdgfra*$^{tm11(EGFP)Sor}$; *Pdfgra*$^{GFP/+}$).

## Surgically induced myocardial infarction

To induce acute MI, mice were anaesthetized by intraperitoneal injection of a combination of ketamine (100 mg/kg) and xylazine (20 mg/kg), and intubated. Hearts were exposed via a left intercostal incision followed by ligation of the left anterior descending coronary artery. Sham operated mice underwent surgical incision without ligation. Hearts were harvested for paraffin-embedding, or FACS analysis at 3 or 7 days post-surgery, as indicated in Results.

## FACS experiments

TIP were isolated as previously described (*Chong et al., 2011*). Briefly, hearts were minced and incubated in collagenase type II (Worthington, USA) at 37°C before filtering through 40 µm strainers. Cells were resuspended in red cell lysis buffer, followed by dead cell removal, immunostaining for 15 min on ice with fluorophore-conjugated antibodies and two times wash with FACS buffer (1x PBS containing 2% fetal bovin serum) before acquisition.

We employed very stringent gating strategies to exclude doublets in the FACS analysis: FSC-H vs FSC-A, FSC-H vs FSC-W and SSC-H vs SSC-W cytograms were used to discriminate and gate out doublets/cell aggregates during sorting or from the analysis (*Figure 1—figure supplement 5*). To account for the autofluorescence generated by MI, we used a wild-type MI mouse as control to set up the gating strategy (*Figure 1—figure supplement 6*).

For the TIP fraction, total DAPI-negative live single cells were sorted in FACS buffer. For the GFP$^+$ fraction, GFP$^+$CD31$^-$cells were sorted from the DAPI-negative live single cells. For Fluidigm experiments, SCA1$^+$PDGFRα$^+$CD31$^-$(S$^+$P$^+$) cells were isolated and sorted as described above. For each sample, at least 10,000 final gate events were collected and stored for the later analyses.

## Colony formation assay

TIP cells were isolated as described and stained with indicated antibodies. FACS sorted primary cells were seeded at a clonogenic density of 50 cells/cm$^2$ (500 cells per well of 6-well plate) and were cultured in α-Minimal Essential Medium (α-MEM) containing 20% FBS+1% Pen/Strep at 37°C in a humidified 2% O$_2$ and 5% CO$_2$ incubator, with medium changes every 2–3 days. After 8-day culture, colonies were rinsed with phosphate-buffered saline (PBS), fixed with 2% paraformaldehyde (PFA) and stained with 0.05% (v/v) crystal violet dye in water. Differences in colony number and size were evaluated by a two-tailed one-sample *t*-test to test for variability between individual samples.

## Sectioning, immunohistochemistry and confocal microscopy

### Paraffin sections

Hearts were fixed in 4% PFA for 24 hr and processed at the Garvan Histopathology Center using a Leica Peloris II - Dual Retort rapid tissue processor (Germany). 10 µm thick longitudinal sections were dewaxed in xylene and rehydrated in decreasing concentrations of ethanol before being washed in distilled water.

### Cryo-sections

Hearts were fixed in 4% PFA for 2.5 hr and washed in PBS before being incubated in 30% w/v Sucrose/PBS overnight at 4°C. Tissues were embedded in Tissue-Tek (Sakura, Cat #4583) and frozen on dry ice. 8 µm thick longitudinal sections were prepared for immunohistochemistry.

### Immunohistochemistry

For paraffin-embedded tissue, antigens were unmasked using Tris-EDTA buffer (Tris base 10 mM, EDTA 1 mM, Tween-20 0.05%) boiled for 20 min. Samples were washed in PBS and treated with 5% BSA, 0.1% Triton X-100 for 1 hr at room temperature. Primary antibodies were incubated overnight at 4°C: chicken anti-GFP (Abcam, Cat #ab13970, 1/200), rabbit anti-WIF1 (Abcam, Cat #ab186845,

1/1000), rat anti-CD31 (Dianova, Cat #DIA-310, Clone S231, 1/100), rat anti-CD45 (BD Biosciences, Cat #553076, Clone 30-F11, 1/100), mouse anti-αSMA (Sigma, Cat #A2547, Clone 1A4, 1/100), rat anti-Ki67 (Dako, Cat #M7249, Clone TEC-3, 1/100) and mouse anti-GM130 (BD Biosciences, Cat #610822, Clone 35/GM130, 1/400). Alexa-conjugated secondary antibodies were incubated 1 hr at room temperature (1/500): goat anti-chicken Alexa 488 (Life Technologies, Cat #A11039), goat anti-rabbit Alexa 555 (Life Technologies, Cat #A21429), goat anti-rabbit Alexa 680 (Life Technologies, Cat #A21109), goat anti-rat Alexa 555 (Life Technologies, Cat #A21434) and donkey anti-mouse Alexa 594 (Life Technologies, Cat #A21203), and counterstained with Hoechst 33258 for 10 min at room temperature (Invitrogen, Cat #H3569, 1/10000). Finally, samples were mounted with ProLong Gold antifade reagent (Invitrogen, Cat #P36934) and imaged using a Zeiss LSM700 upright confocal microscope (Germany).

Images were processed using the ImageJ software (NIH, USA) and a minimum of 2600 cells were quantified in the border zone of 4 infarcted hearts.

## Single-cell transcriptomics platform: 10x Chromium

The single-cell library preparation relied on a commercially available droplet method, the 10x Genomics Chromium System (10x Genomics Inc San Francisco, CA). The number of cells loaded on the system was calculated based on the desired number of captured cells following manufacturer's instructions. scRNA-seq libraries were generated following capture. Chromium GFP$^+$ day 3 samples were sequenced on the Illumina HiSeq 2500 and the remainder on the Illumina NextSeq 500 high output.

## Single cell microfluidics: Fluidigm C1

This experiment was performed using the Fluidigm C1 platform (Fluidigm, San Francisco, CA). Immediately after cell sorting and counting, cells were loaded on the integrated fluidics circuits (IFC) C1 chip. Each capture site was carefully examined under a Leica fluorescence microscope in bright field, Red and Green fluorescence channels for cell doublets and viability, and to ensure the capture rate was satisfactory before cell lysis and cDNA preparation. The reverse transcription was performed on the chip using Clontech SMARTer Ultra Low Input RNA Kit V4 (Takara-Clontech, USA). After running the C1 Fluidigm system, single cell RNA libraries were generated from 100 to 300 pg (picogram) of cDNA, using the low throughput Nextera XT DNA library prep kit (Illumina, USA). Individual barcoded libraries were sequenced by Illumina HiSeq 2500 (125bp).

## Processing of 10x Genomics Chromium scRNA-seq data

Raw scRNA-seq data was processed using the 10x Genomics CellRanger software (version 1.3.0). The BCL files obtained from the Illumina NextSeq platform were processed to Fastq files using the CellRanger *mkfastq* program. The Fastq files were mapped to the mm10 version 1.2.0 reference, downloaded from the 10x Genomics website, with the sequence for H2B-eGFP appended to the reference. The CellRanger *count* program was run on individual Fastq data-sets from the different conditions. The *aggr* program was run to generate aggregate unique molecular identifier (UMI) count matrices for the following experimental data-sets analyzed in this study: (1) TIP: sham-day 3, MI-day 3 and MI-day 7; (2) GFP$^+$: sham-day 3, sham-day 7, MI-day 3 and MI-day 7, (3) GFP$^+$ day 3: sham-day 3 and MI-day 3; (4) GFP$^+$ day 7: sham-day 7 and MI-day 7.

## Filtering, dimensionality reduction and clustering of scRNA-seq data

Bioinformatics processing of the scRNA-seq data was performed in R (*R Development Core Team, 2018*) using the *Seurat* package (*Butler et al., 2018*) with figures primarily generated using ggplot2 (*Wickham, 2009*). R scripts containing the steps used for processing and clustering the data for each individual data-set (TIP aggregate, GFP$^+$ aggregate, GFP$^+$ day 3 and GFP$^+$ day 7) are available in *Source code 1*. For all data-sets, initial quality control filtering metrics were applied as follows. Cells with fewer than 200 detected expressed genes were filtered out. Genes that were expressed in less that 10 cells were filtered out. In order to control for dead or damaged cells, cells with over 5% of raw UMIs mapping to mitochondrial genes were filtered out. To further control for potential doublets in our data-sets, we visualized the distribution of expressed genes and UMI numbers and filtered out cells which were clear outliers.

UMIs were normalized to counts-per-ten-thousand, log-transformed, and a set of highly variable genes was identified by gating for mean expression level and dispersion level in a per-data-set manner. The log-normalized data was scaled, with variation due to total number of UMIs regressed out using a linear model. Principal component analysis was run on the scaled data for the set of previously defined highly variable genes. In order to identify the number of principal components (PCs) to use for clustering, we ran the JackStraw procedure implemented in *Seurat* that identifies statistically significant PCs. Based on running the JackStraw procedure with 1000 permutations, we defined significant PCs as those up to p<0.001, which were used as input to the *Seurat* graph-based clustering program, *FindClusters*. We experimented with modifying the number of PCs used for clustering but found that varying the number of PCs used caused only minor impact on the clustering solutions. The resolution parameter for *FindClusters,* which determined the number of returned clusters, was decided on a per-data-set basis after considering clustering output from a range of resolutions. The cells and clusters were visualized on a t-SNE dimensionality reduction plot generated on the same set of PCs used for clustering.

We inspected the clusters for hybrid gene expression signatures that could indicate captured cell doublets, or a signature of stress/apoptosis that could indicate cells damaged during the process of cell sorting and capture in the microfluidics device. Within TIP, our initial clustering analysis returned 29 clusters; within these we identified five minor clusters exhibiting hybrid gene expression signatures: F-EC, M2MΦ-EC, EC-L1, EC-L2 and BC-TC (*Figure 1—figure supplement 4A–D*). The small size of these populations meant that we could not exclude the possibility of doublets; the five hybrid clusters were therefore removed and all subsequent analysis (e.g. differential expression analysis) was performed on the remaining 24 clusters. Within the GFP$^+$ data-set, we identified one cluster where GO analysis suggested cells in a stressed state; these cells were removed prior to downstream analyses of the GFP$^+$ data-sets.

## Clustering of GFP$^+$ day 7 data-set

We found when clustering the GFP$^+$ day 7 data-set that the clustering solutions, depending on the resolution parameter, tended to either under-cluster (too few clusters) or over-cluster (too many clusters) the data when compared to clustering solutions for the full GFP$^+$ aggregate or GFP$^+$ day 3 data-set. In order to achieve a clustering solution that was directly comparable to the GFP$^+$ aggregate and GFP$^+$ day 3 data, we first over-clustered the data then ran an iterative procedure to 'collapse' transcriptionally similar clusters. We first generated a dendrogram of cluster similarity using the *Seurat BuildClusterTree* program, which builds a phylogenetic tree by first calculating average RNA expression across clusters, then performed hierarchical clustering on a distance matrix calculated from the averaged RNA profiles. We then ran *AssessNodes* to identify the clusters that were the most transcriptionally similar according to the *Random Forest* out-of-bag error calculations. The most similar clusters were merged and the process was repeated. We found four iterations of the above procedure produced clustering results directly comparable to the GFP$^+$ aggregate and GFP$^+$ day 3 clustering solutions.

## Processing and filtering of Fluidigm data

Fastq files were mapped to the Gencode mouse mm10 reference using STAR aligner (*Dobin et al., 2013*) (version 2.5.2a). Reads marked unaligned by STAR were mapped using Bowtie 2 (*Langmead and Salzberg, 2012*) (version 2.3.1) with parameters *–local –very-sensitive-local*. BAM files generated by STAR and Bowtie2 were merged and read counts generated on the merged BAM using the Subread *featureCounts* (*Liao et al., 2014*) program with parameters *-p -T 12 t exon -g gene_name*.

Count data was normalized to counts per-million (CPM) and transformed to Log2(CPM + 1). Percent of RNA mapped to mitochondrial genes per cell was calculated. We filtered out cells that had below 1 million reads or greater than 20% of reads mapped to mitochondrial genes. This yielded 52 cells for experiment 1 and 52 cells for experiment 2.

## Rank-based classifier for comparing data-sets

In order to circumvent the difficulties associated with clustering small numbers of cells, we instead used classification to map cell population identities from our analysis of the Chromium GFP$^+$

experiment to Fluidigm. Due to significant differences in sequencing depth between the experiments, we built an *iterative Random Forest* (iRF) classifier trained on relative gene rankings (i.e. ranking genes in each cell from highest to lowest expressed), hypothesising that using ranks instead of expression should alleviate some of the difficulties in comparing data-sets of different sequencing depths. The classifier was built as follows. As the Fluidigm $S^+P^+$ experiment was performed on healthy hearts, we first removed the injury conditions from the GFP$^+$ data-set and retained the populations most representative of the sham conditions: F-SL, F-Act, F-SH, F-Trans and F-WntX. We re-calculated a set of 706 highly variable genes for GFP$^+$ by gating for genes with higher dispersion and mean expression and took the overlap with expressed genes (CPM > 0 in at least 1 cell) in Fluidigm. The expression matrix of highly variable genes was converted to a rank matrix, which was used as training data for an iRF classifier with 1000 trees and 3 iterations. We evaluated the accuracy of the classifier with 10-fold cross-validation and receiver operating characteristic (ROC) analysis. We found high accuracy across populations with area under the ROC curve (AUC) over 0.95 for all populations (*Figure 4—figure supplement 2A*). The classifier was applied to the Fluidigm data by first converting the expression matrix (highly variable genes) to ranks and returning the most probable cluster assignment. This was done for the two individual Fluidigm experiments (*Figure 4—figure supplement 2B*).

## Comparing GFP$^+$ day 3 and GFP$^+$ day 7 populations

We compared the clusters identified in the GFP$^+$ day 3 to GFP$^+$ day 7 data-sets using two approaches. First was to build a multi-class iRF classifier for the GFP$^+$ day 3 populations as above but including both sham and MI conditions. The classifier was trained on a set of 914 overlapping highly variable genes between the data-sets. We found the iRF classifier maintained reliable prediction accuracy even when including the injury conditions as determined by cross-validation and evaluation with multiple metrics including AUC, sensitivity, specificity and precision (*Figure 8—figure supplement 1B*). The GFP$^+$ day 7 expression matrix was then converted to ranks and the iRF was applied to obtain population probabilities for each cell (*Figure 8—figure supplement 1C*). We additionally counted the number of cells with iRF score >0.5 (*Figure 8—figure supplement 1D*).

## Differential expression

For calculating DE, we first identified genes expressed in at least 25% of cells for at least one of the populations being compared and with an absolute log2 fold-change difference of 0.5 (including a pseudo-count of 1). We then assigned p-values using the 'bimodal' test for DE (*McDavid et al., 2013*) implemented in the *Seurat FindMarkers* program. A Bonferroni-adjusted p-value of 1e-05 was used to determine significantly DE genes.

## Trajectory (Diffusion Map) analysis

Trajectory analysis was performed using *Diffusion Maps* implemented in the *Destiny* R package (*Angerer et al., 2016*). For the set of populations being tested, we took the top upregulated genes for each population and the corresponding log-normalized expression matrix was input to the *DiffusionMap* program with default parameters. A fold-change (log2) cutoff of 1 was used to select upregulated genes with the exception of the GFP$^+$ day 7 inter-MYO analysis, where a cutoff of 0.5 was used. We tested the stability of the *DiffusionMap* output by altering the numbers of input genes (i.e. modulating fold-change cutoff and using the full set of highly variable genes) but found the resulting trajectories to be consistent.

## Gene ontology testing

Over-representation of GO terms in gene lists was calculated using the PANTHER web-service (*Mi et al., 2017*). The set of expressed genes in the relevant experiment was used as background. A false-discovery rate cut-off of 0.05 was used to determine statistical significance.

## Identification of additional minor GFP$^+$ populations

The EC (Endothelial Cell) population, identified by *Pecam1* (CD31) expression, were observed in all conditions (*Figure 4—figure supplement 3*) and their transcriptome showed a strong positive correlation with previously isolated adult cardiac CD31$^+$ ECs (Pearson's correlation test, p-value=3.6e-15,

p=0.48) (*Quaife-Ryan et al., 2017*). Detection of ECs was surprising, given the FACS gates to exclude CD31$^+$ cells. Thus, the EC cluster may have low cell-surface CD31.

The MAC population, unique to MI-day 3, were *Cd45$^+$Cd68$^+$* MΦ (*Figure 4—figure supplement 3*) and likely correspond to the minority population of *Pdgfra*-GFP$^+$CD45$^+$ cells identified at day 5 post-MI (*Asli et al., 2017*). We were also able to identify *Pdgfra*-GFP$^+$CD45$^+$ cells in MI-day 3 samples, as shown in *Figure 6G*. Very few cells in EC and MAC clusters showed GFP expression (*Figure 4E*) and were likely detected because of GFP perdurance. It is unclear at present whether they derive from intra-cardiac or extra-cardiac GFP$^+$ cells.

## Differential proportion analysis

We developed an approach for detecting changes in population proportions across conditions (*Source code 1*). For some number of conditions to be compared, clustering is first performed on an aggregate of all cells across conditions. Cells are assigned two labels: a group (*G*) label representing experimental group/condition and a cluster label (*L*). A count table is generated for each cluster per-condition, which can be converted to a proportion table. We define a statistic for the differential proportion test, $\Delta p_j$, as the difference in cluster proportions between two conditions; that is $\Delta p_j = p_{1j} - p_{2j}$ for some cluster j and corresponding proportions in experiments 1 and 2. This workflow is illustrated in *Figure 1—figure supplement 3A*.

We next construct a null distribution for $\Delta p_j$ by randomly permuting cluster labels *L* for some *w* proportion of *n* total cells. Specifically, *w\*n* cells are randomly selected, and their cell-type labels are replaced by a random sub-sample of labels drawn from the labels from all the cells (sampling without replacement). A new count and proportion table is then generated from this randomized data (*Figure 1—figure supplement 3B*). This process is repeated *t* times, and the resulting $\Delta p_j$ across the randomized data forms the null distribution. We then calculate empirical p-values representing either an increase or decrease in $\Delta p_j$ such that

$$P_{increase} = \frac{1}{t}\sum_{i=1}^{t}\mathbf{I}\left(\Delta p_i \geq \Delta p_j\right)$$

$$P_{decrease} = \frac{1}{t}\sum_{i=1}^{t}\mathbf{I}\left(\Delta p_i \leq \Delta p_j\right)$$

Where I(•) is the indicator function. A final p-value, $P_j$ is determined as the minimum of $P_{increase}$ and $P_{decrease}$. The most important parameter that needs to be set for DPA is *w*, where lower values will trend towards a stricter test (fewer significant hits) and higher values trend towards higher numbers of significant hits. For the following tests, and throughout this paper, we use *w* = 0.1.

As a negative control experiment, we evaluated DPA on our two GFP$^+$ sham experiments (*Figure 1—figure supplement 3C*), which would be expected to demonstrate no major differences in population composition. We compared the results of DPA (*w* = 0.1, *t* = 100,000) to performing Fisher's exact tests. We found Fisher's exact test returned 7/11 of the populations as having significantly different proportions between the two sham experiments with p<0.05 (*Figure 1—figure supplement 3D*). In contrast, DPA identified only one population with significant proportion change between conditions with p=0.03 (*Figure 1—figure supplement 3D*). As p=0.03 represented the most significant change between sham conditions, we used a conservative p-value cutoff of 0.01 for comparing injury time-points as presented in Results.

We further evaluated DPA using simulation testing. We first designed a simulation experiment involving two replicate scRNA-seq experiments performed on one biological system with 10 cells types; that is where the underlying cell proportions are the same in both experiments. We simulated varying degrees of noise (e.g. experimental error or biological variability) by introducing an error rate *e*. Error was introduced when creating a simulated profile of each cell-type per-experiment by randomly adjusting each proportion, *p*, by ±*e\*p* with the proportions finally readjusted to sum to 1. We randomly drew 5000 and 3000 cells from experiment 1 and experiment 2, respectively and performed DPA and Fisher's exact test for each population. The procedure was repeated 100 times with error rates of 0.01, 0.05, 0.1, 0.15 and 0.2. We compared specificity measurements between Fisher's exact test and DPA and found that DPA consistently made fewer false-positive calls than

Fisher's exact test, with difference in specificity between the two methods increasing with higher error rates (*Figure 1—figure supplement 3E*).

We next simulated a control vs condition experiment with 10 cell populations, where six populations change proportionally in the condition and four remain the same. We performed simulated experiments including error as above, drawing 4000 and 6000 cells for the control and condition experiments, respectively. For this experiment we have both true changes (positives) and non-changes (negatives) and could therefore evaluate both true-positive and false-positive detection rates. We found that both Fisher's exact test and DPA correctly identified all true population changes with a sensitivity of 1 across all error rates (*Figure 1—figure supplement 3F*). When considering false-positives, DPA again outperformed Fisher's exact test in specificity and precision measurements (*Figure 1—figure supplement 3G,H*). These results demonstrate that while DPA can identify true proportion changes with comparable sensitivity to Fisher's exact test, it better controls for the detection of false-positive changes.

## Ligand-receptor networks

In order to represent cell-cell communication networks via ligand-receptor interactions, we implemented a directed, weighted network with four layers of nodes as follows. The top layer of nodes refers to a set of *source* cell populations, defined as the cell populations expressing ligands. The second layer of nodes represents the set of ligands expressed by the source populations. A weighted edge connects Source:Ligand where the weight is the Log2 fold-change of the ligand in the source compared to the remaining populations. The third layer of nodes is the receptor targets of the ligands. These are determined from a map of ligand-receptor pairs, collated using both known ligand-receptor interactions and interactions predicted though protein localization and protein-protein interaction (PPI) information (*Ramilowski et al., 2015*). As this map contains human ligand-receptor interactions, we add a weight using mouse-specific protein-protein associations from the STRING database (*Szklarczyk et al., 2017*); these are represented as values between 0 and 1. The fourth layer of nodes represents the *target* cell populations. Receptors are connected to target populations where they are expressed with weight again determined by Log2 fold-change. We did not normalize the three edge weights so as to ensure that gene expression provides the greatest weighting. A *path weight* connecting a source to target via a ligand:receptor pair is calculated as the sum of weights along that path.

For our analysis, we initially considered all ligand:receptor pairs expressed in at least 10% of cells in a population. We then built a network using all 24 sub-populations identified in the TIP data-set. In order to filter out downregulated ligand-receptor connections, we set a minimum path weight of 1.5. An overall weight describing the strength of connection between a source and target population, $w_{s:t}$, could then be calculated by summing all path weights between the source and target.

In order to identify Source:Target connections that have significantly higher summed path weights than would be expected by chance we generated random networks as follows. Given a Source:Target connection we identified the number of total unfiltered paths ($T$), the set of paths with weight greater than 1.5 ($G_{s:t}$) and the subsequent summed path weights ($w_{s:t}$) for $G_{s:t}$. We then generated random networks by retaining the Ligand:Receptor edges (i.e. PPI connections) but randomly selecting $T$ number of Source:Ligand edges and Receptor:Target edges and re-calculating $G_{s:t}$ and $w_{s:t}$ (i.e. sub-sampling the fold-changes). This process was repeated $m$ = 100,000 times and empirical p-values were calculated as

$$P_w = \frac{1}{m}\sum_{i=1}^{m}\mathbf{I}(w_i \geq w_{s:t})$$

As considering all possible combinations of Source:Target paths yields a large number of tests, we adjusted p-values with the Benjamini-Hochberg correction method and considered all edges with adjusted $P_w$ <0.01 to be significant. Significant Source:Target edges were visualized as a hierarchical graph using Cytoscape (*Shannon et al., 2003*) with edge thickness determined by $w_{s:t}$. The code for performing the above analysis on the TIP scRNA-seq is available in *Source code 1*.

## Acknowledgements

We thank Elvira Forte, Gurpreet Kaur (Millennium Science) and Vikram Tallapragada for technical assistance, Rob Salomon, Dominik Kaczorowski and other GWCCG staff for support on FACS and single cell RNA-Seq platforms, and Karen Brennan and Michelle Catwright for animal support.

## Additional information

### Competing interests

Richard P Harvey: Reviewing editor, *eLife*. The other authors declare that no competing interests exist.

### Funding

| Funder | Grant reference number | Author |
|---|---|---|
| University of New South Wales | | Nona Farbehi |
| National Heart Foundation of Australia | 100848 | Joshua WK Ho |
| National Health and Medical Research Council | 1105271 | Joshua WK Ho |
| Stem Cells Australia | SR110001002 | Richard P Harvey |
| National Health and Medical Research Council | 1074386 | Richard P Harvey |
| Fondation Leducq | 15CVD03 | Richard P Harvey |
| St. Vincent's Clinic Foundation | 100711 | Richard P Harvey |
| Fondation Leducq | 13CVD01 | Richard P Harvey |
| National Health and Medical Research Council | 573707 | Richard P Harvey |
| National Health and Medical Research Council | 1118576 | Richard P Harvey |
| New South Wales Cardiovascular Research Network | | Richard P Harvey |

The funders had no role in study design, data collection and interpretation, or the decision to submit the work for publication.

### Author contributions

Nona Farbehi, Ralph Patrick, Conceptualization, Resources, Data curation, Software, Formal analysis, Validation, Investigation, Visualization, Methodology, Writing—original draft, Project administration, Writing—review and editing; Aude Dorison, Conceptualization, Data curation, Funding acquisition, Validation, Investigation, Visualization, Project administration, Writing—review and editing; Munira Xaymardan, Investigation, Writing—review and editing; Vaibhao Janbandhu, Conceptualization, Data curation, Investigation, Visualization, Writing—review and editing; Katharina Wystub-Lis, Conceptualization, Data curation, Supervision, Validation, Investigation; Joshua WK Ho, Conceptualization, Data curation, Software, Formal analysis, Supervision, Funding acquisition, Investigation, Methodology, Writing—review and editing; Robert E Nordon, Conceptualization, Supervision, Funding acquisition, Investigation, Project administration, Writing—review and editing; Richard P Harvey, Conceptualization, Supervision, Funding acquisition, Investigation, Visualization, Writing—original draft, Project administration, Writing—review and editing

### Author ORCIDs

Ralph Patrick [iD] http://orcid.org/0000-0003-0956-1026
Richard P Harvey [iD] http://orcid.org/0000-0002-9950-9792

### Ethics

Animal experimentation: This research was performed following the guidelines, and with the approval, of the Garvan Institute of Medical Research/St. Vincent's Animal Experimentation Ethics Committee (research approvals 13/01, 13/02, 16/03 and 16/10).

### Decision letter and Author response

Decision letter https://doi.org/10.7554/eLife.43882.060
Author response https://doi.org/10.7554/eLife.43882.061

## Additional files

### Supplementary files

• Source code 1. R code for processing and clustering of scRNA-seq data-sets, differential proportion analysis and cell communication network analysis.
DOI: https://doi.org/10.7554/eLife.43882.034

• Supplementary file 1. Differentially expressed genes across TIP sub-populations.
DOI: https://doi.org/10.7554/eLife.43882.035

• Supplementary file 2. Differential proportion analysis p-value results for TIP and GFP+ sub-populations.
DOI: https://doi.org/10.7554/eLife.43882.036

• Supplementary file 3. Differentially expressed genes between Mo/MΦ sub-populations in TIP.
DOI: https://doi.org/10.7554/eLife.43882.037

• Supplementary file 4. Differentially expressed genes across GFP+ sub-populations.
DOI: https://doi.org/10.7554/eLife.43882.038

• Supplementary file 5. Differentially expressed genes across GFP+ Diffusion Map trajectories.
DOI: https://doi.org/10.7554/eLife.43882.039

• Supplementary file 6. GO Biological Process terms associated with GFP+ trajectory differentially expressed genes.
DOI: https://doi.org/10.7554/eLife.43882.040

• Supplementary file 7. Differentially expressed genes from GFP+ day 3 injury response populations.
DOI: https://doi.org/10.7554/eLife.43882.041

• Supplementary file 8. GO Biological Process terms associated with GFP+ day 3 injury response populations according to *Diffusion Map* trajectory: F-Act, F-CI and F-Cyc.
DOI: https://doi.org/10.7554/eLife.43882.042

• Supplementary file 9. Differentially expressed genes between myofibroblast sub-populations in GFP+ day 7 scRNA-seq.
DOI: https://doi.org/10.7554/eLife.43882.043

• Supplementary file 10. GO Biological Process terms associated with myofibroblast sub-populations in GFP+ day 7 scRNA-seq.
DOI: https://doi.org/10.7554/eLife.43882.044

• Supplementary file 11. Spearman correlation test comparisons between TGF-β -treated cardiac fibroblast RNA-seq and GFP+ day 7 sub-populations.
DOI: https://doi.org/10.7554/eLife.43882.045

• Transparent reporting form
DOI: https://doi.org/10.7554/eLife.43882.046

### Data availability

Sequencing data have been deposited in the ArrayExpress database at EMBL-EBI (www.ebi.ac.uk/arrayexpress) under accession codes E-MTAB-7376 and E-MTAB-7365.

The following datasets were generated:

| Author(s) | Year | Dataset title | Dataset URL | Database and Identifier |
|---|---|---|---|---|
| Farbehi N, Patrick R, Dorison A, Xaymardan M, Wystub-Lis K, Janbandhu V, Ho JWK, Nordon RE, Harvey RP | 2018 | Single-cell RNA-seq of mouse cardiac interstitial cells 3 and 7 days after sham or myocardial infarction injury | http://www.ebi.ac.uk/arrayexpress/experiments/E-MTAB-7376 | ArrayExpress database, E-MTAB-7376 |
| Farbehi N, Patrick R, Dorison A, Xaymardan M, Wystub-Lis K, Janbandhu V, Ho JWK, Nordon RE, Harvey RP | 2018 | Single-cell RNA-seq of Pdgfra+/Sca1+/Cd31- mouse cardiac cells | https://www.ebi.ac.uk/arrayexpress/experiments/E-MTAB-7365 | ArrayExpress database, E-MTAB-7365 |

The following previously published datasets were used:

| Author(s) | Year | Dataset title | Dataset URL | Database and Identifier |
|---|---|---|---|---|
| Schafer S, Viswanathan S, Widjaja AA | 2017 | Integrated target discovery screens identify IL11 as novel therapeutic target for fibrosis | https://www.ncbi.nlm.nih.gov/geo/query/acc.cgi?acc=GSE97117 | Gene Expression Omnibus, GSE97117 |
| Skelly DA, Squiers GT, McLellan MA, Bolisetty MT, Robson P, Rosenthal NA, Pinto AR | 2017 | Single cell RNA-Seq of the murine non-myocyte cardiac cellulome | https://www.ebi.ac.uk/arrayexpress/experiments/E-MTAB-6173/ | ArrayExpress database, E-MTAB-6173 |
| Quaife-Ryan GA, Sim CB, Ziemann M, Kaspi A | 2017 | Multicellular Transcriptional Analysis of Mammalian Heart Regeneration | https://www.ncbi.nlm.nih.gov/geo/query/acc.cgi?acc=GSE95755 | Gene Expression Omnibus, GSE95755 |
| Bochmann L, Sarathchandra P, Mori F, Lara-Pezzi E, Lazzaro D | 2010 | Transcription profiling of mouse cardiac muscle and epicardium after left coronary artery ligation and sharm operation | https://www.ebi.ac.uk/arrayexpress/experiments/E-MEXP-2446/ | ArrayExpress database, E-MEXP-2446 |

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
