## [Decision Letter]

Thank you for submitting your article "Single-cell expression profiling reveals dynamic flux of cardiac stromal, vascular and immune cells in health and injury" for consideration by *eLife*. Your article has been reviewed by three peer reviewers, one of whom is a member of our Board of Reviewing Editors, and the evaluation has been overseen by Harry Dietz as the Senior Editor. The reviewers have opted to remain anonymous.

The reviewers have discussed the reviews with one another and the Reviewing Editor has drafted this decision to help you prepare a revised submission.

All of the reviewers as well as the editor believe that your study has merit. The reviewers felt that a better synopsis of the informatic analysis that would both simplify and better illustrate the findings would be beneficial to the report. There were also multiple instances where additional data including gene/protein expression, positional information of cell types identified, and confirmation of gene expression differences reported in the scRNA-seq analysis are required. Additional textual revisions are also requested and outlined below.

Reviewer #1:

The authors report on a new fibroblast type/state which is enriched for Wnt inhibition through increased expression of Wnt inhibitors, in particular Wif1. While the data is undoubtedly important and useful for the cardiac repair field, there are several notable issues that limit the impact of this report.

1) The data analysis as presented is confusing and does not lend itself to easy interpretation. If one of the main points to focus on is the Wntx cell type/state, then this should be much more clearly presented in the figures. This should include a clear assessment by GO category or other statistically significant methods to show that this cell cluster is enriched in expression of Wnt inhibitors.

2) In relation to point 1, if the Wntx cell type is enriched in expression of secreted Wnt inhibitors, then these cells could act on a neighboring cell type rather than in a cell autonomous manner. The authors should provide data on whether the Wntx cell is located in any specific relationship to other cardiac cells types. The current data in Figure 5 are at too low a resolution to define cell-cell relationships as Wif1 could be acting at short range.

3) A weakness of these studies is that there is no clear integration across the 3 different time frames examined i.e. 0, 3, and 7 days. To assess true relationships and how cells change after injury, a pseudotime analysis combining the 3 times points could help define what basal homeostatic fibroblast population gives rise to the Wntx and other cell types found at days 3 and 7.

Reviewer #2:

This is a well-written paper that concentrates on delineating the distinct immune and fibrotic signatures present in the acute and reparative phases post cardiac injury. The data presented here give insight into the different subtypes of immune and fibrotic cells in the heart, which have otherwise been hard to identify in bulk cellular and transcriptomic studies. Although this is a largely descriptive study, it provides a wealth of information that will be useful in the future to garner mechanistic insights into the signaling networks involved during cardiac injury and repair.

1) The authors mention the presence of a small percentage of hybrid endothelial cells which display both endothelial and fibroblast markers. To exclude the possibility that this population arises due to captured cell doublets, the authors should perform co-immunostaining for these markers to determine if this intermediate cell type exists in the heart. Immunostaining will also provide important spatial information regarding the location of this cell type in the heart.

2) Similarly, the authors mention that a significant proportion of the M2-macrophage population expressed endothelial markers, suggesting their ability to transdifferentiate into endothelial cells. This conclusion would also be more compelling if coupled with some spatial information. Are these cells prevalent in the areas near the site of injury?

3) In the differential proportion analysis in Figure 1E, the authors should indicate the changes in the total fibroblast population along with the subtypes already shown. There appears to be a significant decrease in the F-SH and F-SL populations at MI-day3 and this not very clearly indicated. In the latter half of the paper, the authors attribute this to the conversion of these fibroblast types to F-Act. However, the FACS data presented in the supplement do not entirely support this conclusion. This should be addressed.

4) Parts of the Discussion section simply reiterate the results. It might be more useful if the provided a more in-depth discussion regarding the implications of the different immune signatures observed post injury in the context of the known literature.

5) The authors need to cite and discuss in some detail the recently published, related findings of Epelman and coworkers "Self-renewing resident cardiac macrophages limit adverse remodeling following myocardial infarction" Nature Immunology 20, 29-39 (2019).

Reviewer #3:

The authors have performed an interesting single cell RNA-seq using the widely used 10X platform on adult ventricular murine cardiac tissue with or without myocardial infarction (MI). The authors focused their study on non-cardiomyocytes (TIP cells) with an emphasis on the fibroblast lineage. Moreover, they extensively describe a population of fibroblasts that express high levels of Wnt signaling-associated transcripts like Wif1.

Overall, while the work is very solid and interesting, it would be improved by further validation experiments and functional data to support the profiling data, which is predictive. The data analysis is of good quality, however, there are several technical issues as discussed below.

1) The authors include and present what appear to be obvious cell doublets in their final data set as EC-L1, EC-L2, and F-EC clusters (Figure 1). Doublets are expected on the 10X platform at a certain percentage which can be as high as ~5%, depending on the number of cells loaded. Without validation, these clusters that are suspected to be doublets should be removed and information regarding their identities and reasons for removal be added to the Materials and methods section. Moreover, the M2MΦ cluster which "expressed canonical endothelial markers" is partly composed of MΦ-endothelial cell doublets. This is obvious given that the endothelial-like "~29%" separates clearly from the main myeloid cluster (Figure 1D). The clustering parameter therefore is inadequate and unable to separate this issue and should be corrected. Finally, suggestions of a connection to trans-differentiation of myeloid cells into the endothelial cells and the contributions of embryonic EMPs to cardiac vessels should be removed from the manuscript to avoid confusion or more experimental data must be presented.

2) Based on the methods described in this study, it is unclear why epicardial and endocardial cells were not detected in the analysis shown in Figure 1. Even single nucleus RNA-seq studies, which are much less powerful, can detect these rare cell types in the heart (Hu et al., 2018). Please clarify in the writing.

3) The authors use the M1 and M2 classifications of macrophages and monocytes throughout their manuscript. Recent publications (that have been accurately referenced in the manuscript) have detailed the transcriptional responses of myeloid cells after MI (King et al., 2017) and after ischemia-reperfusion (IR) injury (Bajpai et al., 2018) using high throughput single cell RNA-sequencing. The authors should consider using nomenclature conventions more consistent with these previous reports.

4) Results from ligand-receptor analysis would greatly benefit from validation at the protein level.

5) It is unclear why the authors performed additional scRNA-seq on Pdgfra lineage cells, and what information was gained from these experiments. Please clarify in the writing. Similarly, what were the findings from the Fluidigm scRNA-seq experiments? Did they differ from or confirm the droplet-based scRNA-seq results? Please clarify in the writing.

6) It is unclear if the authors are claiming that the F-SH cells, which they additionally denote as CFU-Fs are cardiac mesenchymal stem cells (MSCs)? This section of the Results needs to be more clearly written.

7) In Figure 5, no strong evidence is provided that the indicated images are in the borderzone. Please clarify.

---

## [Author Response]

Reviewer #1:The authors report on a new fibroblast type/state which is enriched for Wnt inhibition through increased expression of Wnt inhibitors, in particular Wif1. While the data is undoubtedly important and useful for the cardiac repair field, there are several notable issues that limit the impact of this report.1) The data analysis as presented is confusing and does not lend itself to easy interpretation. If one of the main points to focus on is the Wntx cell type/state, then this should be much more clearly presented in the figures. This should include a clear assessment by GO category or other statistically significant methods to show that this cell cluster is enriched in expression of Wnt inhibitors.

Thank you for this suggestion – we have accordingly made changes to both the figure presentation and structure of the text to more clearly present our findings on F-WntX, and to aid in interpretation of the data. Regarding figures, we have built a new figure (Figure 5) and modified Figure 6 so that they focus exclusively on the F-WntX cells. Figure 5 focusses specifically on the presentation of results from the bioinformatics analysis of the F-WntX population. This includes representative GO terms over-represented in genes up-regulated in F-WntX compared to other populations, including the term *negative regulation of Wnt signalling* (Figure 5C). In relation to comment 2 of reviewer 1, we have incorporated additional figure panels in Figure 5 focussing on the paracrine functions of F-WntX using bioinformatic analysis of ligand-receptor connections (Figure 5D, E). As also requested, we have included high-resolution immunofluorescence images to examine the spatial relationship of F-WntX cells to other cell types (Figure 6F-H).

In the text, we have accordingly re-structured the section on analysis of Pdgfra-GFP^+^ fibroblasts to better streamline the presentation of results. All analyses comparing the F-SL and F-SH populations are now presented together. Similarly, all bioinformatics analysis on F-WntX is now presented within the same section, in relation to Figure 5, followed by the analysis of WIF1 IF in relation to Figure 6. These are presented in sections Novel Pdgfra-GFP^+^ fibroblast populations and Localization and composition of WIF1^+^ cells, respectively. To further aid data interpretation, we have built a schematic of our interpretation of the flux and differentiation/pseudotime dynamics of the GFP^+^ populations (Figure 9), which is referred to in the Discussion.

2) In relation to point 1, if the Wntx cell type is enriched in expression of secreted Wnt inhibitors, then these cells could act on a neighboring cell type rather than in a cell autonomous manner. The authors should provide data on whether the Wntx cell is located in any specific relationship to other cardiac cells types. The current data in Figure 5 are at too low a resolution to define cell-cell relationships as Wif1 could be acting at short range.

As indicated in our response to Comment 1, we have performed additional co-immunostainings and have taken higher resolution images to investigate whether F-WntX cells can be observed to co-localise with specific cell types. These higher-resolution images are contained in Figure 6F-H. Based on 2D distance, Figure 6H shows frequent close proximity or contact between WIF1^+^ cells and endothelial cells marked by CD31. This result suggests a potential interaction between these two cells types which is also suggested by the cell-cell ligand-receptor connections. Such interactions were less evident for WIF1^+^ cells and those expressing α-SMA or CD45 (Figure 6F, G), which is consistent with the lack of a predicted communication between F-WntX cells and MYO or immune cells in the cell communication analysis.

While we hope we have addressed reviewer 1’s concerns with respect to the resolution of immunofluorescence micrographs, we acknowledge that the spatial data presented in not quantitative or 3D. Cardiac tissue is complex, being composed of large cardiomyocytes embedded within an extracellular matrix containing populations of endothelial cells, perivascular cells, fibroblasts, and immune, neural and other cell types, all in close proximity. These cells will be communicating with others via membrane contact (possibly though long membrane protrusions or nanotubes) or gradients of secreted proteins. With MI, tissue structure is degraded and in certain regions surviving cells are overwhelmed by invading immune cells during MI, making it harder to make sense of which cardiac cells are in closest proximity. This complexity precludes a more detailed analysis of the spatial and functional relationship between WIF1^+^ cells and other cardiac interstitial cells for this paper. These questions could be addressed in part using quantitative 3D imaging and a combination of cell type specific markers, and membrane and signalling markers, but we feel that such a high level of experimentation is outside the scope of this work.

We have modified the text in the Results section under Localization and composition of WIF1^+^ cells, and in the Discussion, to reflect the above observations on 2D proximity.

3) A weakness of these studies is that there is no clear integration across the 3 different time frames examined i.e. 0, 3, and 7 days. To assess true relationships and how cells change after injury, a pseudotime analysis combining the 3 times points could help define what basal homeostatic fibroblast population gives rise to the Wntx and other cell types found at days 3 and 7.

Individual tSNE plots representing sham and MI-days 3 and 7 data for TIP and *Pdgfra*-GFP+ populations have been presented at the beginning of each section (Figures 1A, 4A, 8A, B). Furthermore, population proportion data (Figures 1A and 4B; Figure 1—figure supplement 1A, B; Figure 1—figure supplement 4C; Figure 4—figure supplement 1F), population gene expression data (Figure 2C; Figure 1—figure supplement 3D) and cell trajectory analyses (Figures 7B, 8C, D) have been broken down according to sham and MI-days 3 and 7, as appropriate for interpretation. However, in order to manage the large amounts of subsequent data analyses (mostly gene expression data displayed on tSNE plots), we have generally had to display outputs on aggregate plots of sham/MI data. This includes the Diffusion Map presented in Figure 7A, which is nonetheless useful in that is indicates the proposed trajectory of cells from unactivated populations F-SL and F-SH through to F-WntX, F-Act, proliferating cells, and myofibroblasts. These can be interpreted by back-reference to the original tSNE populations plots showing the breakdown according to sham and MI-days 3 and 7 data. Thus, showing all sham/MI data would be unmanageable. However, we understand the reviewer’s point, and so to make data more accessible we have added population labels to some of the aggregate plots. Furthermore, our interpretation of these dynamics is now summarised in the schematic in Figure 9.

Reviewer #2:This is a well-written paper that concentrates on delineating the distinct immune and fibrotic signatures present in the acute and reparative phases post cardiac injury. The data presented here give insight into the different subtypes of immune and fibrotic cells in the heart, which have otherwise been hard to identify in bulk cellular and transcriptomic studies. Although this is a largely descriptive study, it provides a wealth of information that will be useful in the future to garner mechanistic insights into the signaling networks involved during cardiac injury and repair.1) The authors mention the presence of a small percentage of hybrid endothelial cells which display both endothelial and fibroblast markers. To exclude the possibility that this population arises due to captured cell doublets, the authors should perform co-immunostaining for these markers to determine if this intermediate cell type exists in the heart. Immunostaining will also provide important spatial information regarding the location of this cell type in the heart.2) Similarly, the authors mention that a significant proportion of the M2-macrophage population expressed endothelial markers, suggesting their ability to transdifferentiate into endothelial cells. This conclusion would also be more compelling if coupled with some spatial information. Are these cells prevalent in the areas near the site of injury?

We thank the reviewer for the comments regarding doublets, which is an issue that was also raised by reviewer 3 in their Comment 1. In line with reviewer 3’s request that “Without validation, these clusters that are suspected to be doublets should be removed and information regarding their identities and reasons for removal be added to the Materials and methods section”, we have restructured the figures and text. First, we removed all “hybrid” populations from the initial tSNE plots and from subsequent data analyses to produce dendrograms, DE gene lists and GO terms etc, with an explanation given in the text and Materials and methods. However, we recognise the potential biological importance of true hybrid populations as they could betray differentiation or trans-differentiation events that are otherwise difficult to detect. Thus, in response to reviewer 2’s suggestion, we have dedicated a section at the end of the Discussion of TIP cell populations (Hybrid populations in TIP) where we acknowledge the problems of doublets and then present new data in support of two of the hybrid populations identified by scRNAseq. We performed flow cytometry experiments with stringent filtering to exclude doublets and aggregates (see also response to reviewer 3, Comment 1), which shows a minor population of *Pdgfra*-GFP^+^/CD31^+^ cells in both sham and MI hearts, providing support for the presence of F-EC. We also performed flow and detected a population of MΦ cells (identified as CD45^+^F4/80^+^CD206^+^) expressing a recognised marker of ECs (CD31), in support of the M2MΦ-EC population. These data are presented in Figure 1—figure supplements4-7. We acknowledge that we cannot rule out that doublets may still contribute to the hybrid F-EC and M2MΦ-EC populations; however, we believe that our new data warrant a specific discussion of hybrids in this manuscript to highlight the importance of further investigation.

3) In the differential proportion analysis in Figure 1E, the authors should indicate the changes in the total fibroblast population along with the subtypes already shown. There appears to be a significant decrease in the F-SH and F-SL populations at MI-day3 and this not very clearly indicated. In the latter half of the paper, the authors attribute this to the conversion of these fibroblast types to F-Act. However, the FACS data presented in the supplement do not entirely support this conclusion. This should be addressed.

These are good suggestions and we have included an additional supplementary figure panel (Figure 1—figure supplement 1C) that contains the proportion changes of the high-level cell types, including fibroblasts, by merging appropriate clusters (F-SL, F-SH, F-WntX and F-Act into fibroblasts, the EC1, EC2 and EC3 clusters into endothelial cells etc.). We have modified the text in the sub-section on DPA (Differential proportion analysis) to specifically mention the significant decrease in F-SH and F-SL proportions. As we focus more specifically on F-SL and F-SH, including their flux after injury in the section on the Pdgfra-GFP^+^ enriched cells, we also direct readers to this section for a more detailed description of the populations, including their flux after MI. We acknowledge that our flow cytometry data does not prove that the F-SL and F-SH populations definitely differentiate into F-Act, and we have toned down the claim that F-SL/F-SH diminishment occurs as a result of F-Act expansion.

4) Parts of the Discussion section simply reiterate the results. It might be more useful if the provided a more in-depth discussion regarding the implications of the different immune signatures observed post injury in the context of the known literature.

Thank you for this appropriate suggestion. We have substantially shortened the Discussion to avoid too much reiteration of results while retaining pertinent discussion of implications. We have also inserted a longer treatment of monocyte/macrophage populations.

5) The authors need to cite and discuss in some detail the recently published, related findings of Epelman and coworkers "Self-renewing resident cardiac macrophages limit adverse remodeling following myocardial infarction" Nature Immunology 20, 29-39 (2019).

We thank the reviewer for pointing out this recent detailed paper and we have now cited it extensively in the Results and Discussion. Specifically, we have compared the markers for the cardiac resident macrophage/DC sub-populations that were identified by Epelman and co-workers with our data. There is limited scope for a direct comparison as Epelman and co-workers enrich for macrophage/DC populations in their paper using cell surface markers and therefore have more power to identify resident macrophage sub-populations than we do within total cardiac interstitial cells, where tissue-resident macrophages represent only ~3% of cells in sham hearts and ~1% of cells in MI hearts. Pleasingly, however, we were able to discern both the Timd4^-^Lyve1^-^MHC-II^high^Ccr2^-^ and Timd4^+^Lyve1^+^MHC-II^low^Ccr2^-^ self-renewing tissue-resident sub-populations that Epelman et al. describe as the major sub-sets of our tissue-resident macrophage population (MAC-TR). We have therefore cited the Epelman paper in the context of these observations, and have included the expression of the discriminating markers as found in our data – *Timd4* and *Lyve1* in Figure 2—figure supplement 1A, and *MHC-II* and *Ccr2* in Figure 2. We have inserted this into the sub-section, Monocyte/macrophage cell identities and dynamics, as follows:

“Recent work using flow cytometry and scRNA-seq has delineated several subsets of cardiac tissue-resident MΦ, including a self-renewing and pro-regenerative population with the signature *TIMD4*^+^*LYVE1*^+^*MHC-II*^low^*CCR2*^-^, that self-renew and are not replaced by blood monocytes even after injury [Lavine et al., 2018]. […] The additional major subset of CCR2^-^ tissue-resident MΦ [Lavine et al., 2018 could also be recognised at the scRNAseq level as the *Timd4*^-^*Lyve1*^-^*H2-Aa(MHC-II*)^high^*Ccr2*^-^ subset of MAC-TR – this population has been shown to have a low monocyte dependence during homeostasis but is almost fully replaced by monocytes after MI [Lavine et al., 2018]. “

Reviewer #3:The authors have performed an interesting single cell RNA-seq using the widely used 10X platform on adult ventricular murine cardiac tissue with or without myocardial infarction (MI). The authors focused their study on non-cardiomyocytes (TIP cells) with an emphasis on the fibroblast lineage. Moreover, they extensively describe a population of fibroblasts that express high levels of Wnt signaling-associated transcripts like Wif1.Overall, while the work is very solid and interesting, it would be improved by further validation experiments and functional data to support the profiling data, which is predictive. The data analysis is of good quality, however, there are several technical issues as discussed below.1) The authors include and present what appear to be obvious cell doublets in their final data set as EC-L1, EC-L2, and F-EC clusters (Figure 1). Doublets are expected on the 10X platform at a certain percentage which can be as high as ~5%, depending on the number of cells loaded. Without validation, these clusters that are suspected to be doublets should be removed and information regarding their identities and reasons for removal be added to the Materials and methods section. Moreover, the M2MΦ cluster which "expressed canonical endothelial markers" is partly composed of MΦ-endothelial cell doublets. This is obvious given that the endothelial-like "~29%" separates clearly from the main myeloid cluster (Figure 1D). The clustering parameter therefore is inadequate and unable to separate this issue and should be corrected. Finally, suggestions of a connection to trans-differentiation of myeloid cells into the endothelial cells and the contributions of embryonic EMPs to cardiac vessels should be removed from the manuscript to avoid confusion or more experimental data must be presented.

See also Comments 1 and 2 of reviewer 2. We thank the reviewers for their insightful suggestions and we have performed a number of additional analyses to address these concerns. As reviewer 3 suggests, we have increased the clustering resolution so that the hybrid M2-macrophage-EC population (M2MΦ-EC) separates as a distinct cluster from the remaining M2-macrophage subset (Figure 1—figure supplement 4A-D). This adjustment did not substantially change the discrimination of other populations – the result was 29 clusters in total, 5 of which we identified as minor hybrid populations including the previously identified EC-L1, EC-L2, F-EC and M2MΦ-EC cells, and a new hybrid population expressing B-cell and T-cell markers (BC-TC). As the reviewer also suggests, we have removed hybrid clusters from the analysis of populations presented in the manuscript (Materials and methods).

Furthermore, as indicated in our response to reviewer 2 Comments 1 and 2, we have performed additional FACS experiments following a stringent process of doublet removal (see Figure 1—figure supplement 5) to test whether we could provide support for hybrid populations F-EC and M2MΦ-EC (described in detail in Hybrid populations in TIP; Figure 1—figure supplement 6, 7). These FACS experiments indeed revealed a GFP^+^CD31^+^ population in MI hearts (Figure 1—figure supplement 6) and a population of CD31^+^ cells co-expressing CD45 and macrophage markers (Figure 1—figure supplement 7). The additional data provide evidence for the existence of these hybrid populations. We have included an additional Results sub-section – Hybrid populations in TIP – where we describe the hybrid populations detected, raise the issue of doublets, and present the supporting FACS data for the 2 populations.

2) Based on the methods described in this study, it is unclear why epicardial and endocardial cells were not detected in the analysis shown in figure 1. Even single nucleus RNA-seq studies, which are much less powerful, can detect these rare cell types in the heart (Hu et al., 2018). Please clarify in the writing.

The Hu et al. study using single nucleus (sn)RNAseq identified a minor (1%) population of cells as epicardial, but did not identify endocardial cells. Skelly et al., 2018, performed scRNAseq using the 10X Chromium platform on interstitial cells of healthy adult murine hearts, but did not identify epicardial or endocardial cells. As commented on by Hu et al., it is possible that snRNA-seq, which overcomes issues of cell dissociation, is better suited than scRNA-seq for obtaining rare populations such as epicardial cells. We have clarified this point in the text in reference to the two above mentioned papers, within the section discussing our detection of epicardial-derived cells among the *Pdgfra*-GFP^+^ population, as follows:

“Consistent with a previous cardiac scRNA-seq analysis [Skelly et al., 2018], we did not detect epicardial cells in TIP data, suggesting that these cells are under-sampled in our experiments – this is likely technical as epicardial cells have been detected readily in single-nucleus RNA-seq [Hu et al., 2018].”

3) The authors use the M1 and M2 classifications of macrophages and monocytes throughout their manuscript. Recent publications (that have been accurately referenced in the manuscript) have detailed the transcriptional responses of myeloid cells after MI (King et al., 2017) and after ischemia-reperfusion (IR) injury (Bajpai et al., 2018) using high throughput single cell RNA-sequencing. The authors should consider using nomenclature conventions more consistent with these previous reports.

We thank the reviewer for their suggestion. We have endeavoured where possible to use nomenclature consistent with the previous papers. Regarding King et al. we have adopted their nomenclature for IFN-inducible cells to define the corresponding population of myeloid cells in our data that express interferon-responsive genes (MAC-IFNIC population). Regarding Bajpai et al., the focus of their analysis was on resident macrophage populations, and as we noted in our response above to reviewer 2 Comment 5, we do not have sufficient representation of tissue-resident macrophages to identify the many sub-types of myeloid cells that they report in their study. We believe that the nomenclature we are using is suitable for the data and analysis contained within our manuscript. At every stage, we provide the defining gene expression signature for each population identified.

4) Results from ligand-receptor analysis would greatly benefit from validation at the protein level.

We thank the reviewer for the suggestion. As indicated in our response to reviewer 1 Comment 2, we have performed additional co-staining experiments to provide support for the ligand-receptor analysis. In one experiment, we look at the relationship between WIF1+ cells and neighbouring cell types (see response to reviewer 1 Comment 2 for details and revised Figure 6). In an additional analysis, we looked at the relationship between GFP^+^ fibroblasts and CD31^+^ endothelial cells as the ligand-receptor map predicted a significant paracrine relationship between fibroblast populations and endothelial cells. Our co-stainings showed that GFP^+^ cells were indeed frequently observed in close spatial relationship to or in contact with CD31^+^ cells (shown in Figure 3—figure supplement 1A-B), with the additional finding the GFP and CD31 never overlap. We have added the following description to the sub-section Cell-cell communication analysis in TIP:

“Strikingly, fibroblast populations (F-SH, F-SL, F-Act and F-Wntx) appeared to communicate exclusively with vascular (ECs and mural) and glial cells. In line with this result, immunofluorescence analysis of sham and MI-day 3 hearts showed that *Pdgfra*-GFP^+^ fibroblasts were observed in close spatial relationships or direct contact with CD31^+^ endothelial cells (Figure 3—figure supplement 1A-B).”

5) It is unclear why the authors performed additional scRNA-seq on Pdgfra lineage cells, and what information was gained from these experiments. Please clarify in the writing. Similarly, what were the findings from the Fluidigm scRNA-seq experiments? Did they differ from or confirm the droplet-based scRNA-seq results? Please clarify in the writing.

The *Pdgfra*-GFP^+^ experiment using the 10X Chromium platform allowed us to enrich for fibroblasts at MI time-points that are otherwise dominated by the presence of immune cells, which dilute out the fibroblast sub-types. For example, at MI-day 3, over 80% of cells were myeloid in our data (Figure 1—figure supplement 2A), severely limiting observations we could made about fibroblasts. In fact, we gained significant new insights into fibroblast subsets using the enrichment approach. The discrimination of F-Act, F-Cyc and F-CI could not have been achieved using the TIP data-set alone. Likewise, by enriching for *Pdgfra*-lineage cells at MI-day 7, we were able to identify for the first time myofibroblast sub-types with contrasting fibrotic and anti-fibrotic signatures. We have clarified this point in the sub-section Single-cell RNA-seq of the Pdgfra-GFP^+^ cardiac fibroblast lineage as follows:

“To circumvent the dominance of immune cells in TIP following MI, which dilute out other cell populations, and to focus on fibroblast sub-populations (Figure 1A), we performed single-cell expression profiling on PDGFRα^+^CD31^-^ cardiac interstitial cells at days 3 and 7 post-sham or MI.”

The Fluidigm experiment was performed on FACS-sorted SCA1^+^/PDGFRA^+^/CD31^-^ (S^+^P^+^) cells that we had previously shown contained fibroblast colony-forming cells (cCFU-F). The main aims therefore were to connect the 10X Chromium data with this subset and to provide a deeper sequencing resource of the cell-types within this fraction. Specifically, it allowed us to test the hypothesis that the F-SH population defined from the 10X Chromium scRNA-seq data corresponded in large part to S^+^P^+^. The small number of cells profiled on Fluidigm (two independent experiments, ~50 cells each) meant that we could not perform a meaningful unbiased clustering analysis, as with the Chromium scRNA-seq. Instead, we used a machine learning classification approach (Materials and methods: Rank-based classifier for comparing data-sets) allowing us to demonstrate that the F-SH cells are predicted to correspond to the majority of cells from the FACS-defined S^+^P^+^ fraction. This leads to a very important conclusion: that the main fibroblast populations, F-SH and F-SL, have different functional properties with respect to colony formation. We have modified the section in Single-cell RNA-seq of the *Pdgfra*-GFP^+^ cardiac fibroblast lineage to clarify the purpose of the Fluidigm experiment, as follows:

“In order to confirm the relationship between F-SH and S^+^P^+^, we performed deeper scRNA-seq on 103 FACS-purified S^+^P^+^ cells from uninjured wild type mice using the *Fluidigm* platform and predicted cell identity using an *iterative Random Forest* (iRF) classifier [Basu et al., 2018] trained on populations defined in our GFP^+^ experiments in sham conditions using the *Chromium* platform (Materials and methods; Figure 4—figure supplement 2A).”

6) It is unclear if the authors are claiming that the F-SH cells, which they additionally denote as CFU-Fs are cardiac mesenchymal stem cells (MSCs)? This section of the Results needs to be more clearly written.

We apologise for the confusion. To clarify, we claim that cCFU-F are enriched in the F-SH population – we are *not* claiming that all of F-SH corresponds to cCFU-F or MSCs.

Out claim is based on several findings. First, previous observation based on FACS fractionation of cardiac fibroblasts and clonal colony formation assays allowed us to show that colony-forming cells (cCFU-F) were enriched within a Sca1^+^Pdgfra^+^CD31^-^ (S^+^P^+^) fraction compared to the P+S- fraction (Chong et al., 2011; see also Figure 4 —figure supplement 1A, B). Second, from the analysis of our Fluidigm scRNA-seq of FACS-sorted S^+^P^+^ cells, we show in the current manuscript that the majority of S^+^P^+^ cells are predicted to correspond to the F-SH population we identified from the 10X Chromium scRNA-seq data. As F-SH carries the majority of S^+^P^+^ cells, the implication is that F-SH also carries the majority of the cCFU-F. We have modified the text in this section to make our claim clearer:

“Together, these results show that the F-SH population contains a subset of cells expressing *Pdgfra, Ly6a(Sca1*) and *Thy1(Cd90*) that is enriched in cCFU-F, highlighting the distinct expression signatures and functional properties of F-SH and F-SL.”

7) In figure 5, no strong evidence is provided that the indicated images are in the borderzone. Please clarify.

We have included a new panel (Figure 6A) showing the clear morphological distinction between healthy and infarcted area in the border zone. This image shows the presence of WIF1^+^ cells in the border zone of the damaged tissue and their absence in healthy (“remote”) myocardium.